# Rhodopsin-positive cell production by intravitreal injection of small molecule compounds in mouse models of retinal degeneration

Yuya Fujii[1], Mitsuru Arima🔘[1,2]*, Yusuke Murakami🔘[1], Koh-Hei Sonoda[1]

**1** Department of Ophthalmology, Graduate School of Medical Sciences, Kyushu University, Fukuoka, Japan,
**2** Center for Clinical and Translational Research, Kyushu University Hospital, Fukuoka, Japan

* marima4802@gmail.com

**Data Availability Statement:** All relevant data are within the manuscript and its Supporting Information files.

## Abstract

We aimed to verify whether the intravitreal injection of small molecule compounds alone can create photoreceptor cells in mouse models of retinal degeneration. Primary cultured mouse Müller cells were stimulated *in vitro* with combinations of candidate compounds and the rhodopsin expression was measured on day 7 using polymerase chain reaction and immunostaining. We used 6-week-old N-methyl-N-nitrosourea-treated and 4-week-old rd10 mice as representative *in vivo* models of retinal degeneration. The optimal combination of compounds selected via *in vitro* screening was injected into the vitreous and the changes in rhodopsin expression were investigated on day 7 using polymerase chain reaction and immunostaining. The origin of rhodopsin-positive cells was also analyzed via lineage tracing and the recovery of retinal function was assessed using electroretinography. The *in vitro* mRNA expression of rhodopsin in Müller cells increased 30-fold, and 25% of the Müller cells expressed rhodopsin protein 7 days after stimulation with a combination of 4 compounds: transforming growth factor-β inhibitor, bone morphogenetic protein inhibitor, glycogen synthase kinase 3 inhibitor, and γ-secretase inhibitor. The *in vivo* rhodopsin mRNA expression and the number of rhodopsin-positive cells in the outer retina were significantly increased on day 7 after the intravitreal injection of these 4 compounds in both N-methyl-N-nitrosourea-treated and rd10 mice. Lineage tracing in td-Tomato mice treated with N-methyl-N-nitrosourea suggested that the rhodopsin-positive cells originated from endogenous Müller cells, accompanied with the recovery of the rhodopsin-derived scotopic function. It was suggested that rhodopsin-positive cells generated by compound stimulation contributes to the recovery of retinal function impaired by degeneration.

## Introduction

Various retinal diseases damage the photoreceptor cells, leading to irreversible visual dysfunction [1, 2]. Many studies on induced pluripotent stem cell transplantation and gene therapy to treat the visual dysfunction of retinitis pigmentosa (RP) and age-related macular degeneration

**Funding:** This study was supported by JSPS KAKENHI [grants number JP18H02956 and JP21H03094] given to KH Sonoda. Research funding was acquired from Senju Pharmaceutical Co., Ltd., given to M Arima, Y Murakami, and KH Sonoda. The funders had no role in study design, data collection and analysis, decision to publish, or preparation of the manuscript.

**Competing interests:** The authors have declared that no competing interests exist.

(AMD), which are typical retinal degenerative diseases, have been published in the last decade [3–5]. The results of these preclinical and clinical studies revealed that stem cell therapy [6] and viral gene transfer [7] are effective in restoring vision. However, these therapies have disadvantages, namely, the high cost of developing and manufacturing cellular medicine, the risk of complications, such as immune rejection and tumorigenesis, and the need for special techniques for transplanting cells/cell sheets and vectors [8].

Cell and gene therapies are expected to be widely used in clinical practice in the future. However, they are highly invasive to the patients because they need to inject cells/cell sheets, or solution into the subretinal space. Therefore, only patients who have reached social blindness are eligible for these therapies [6, 7]. Since retinal degenerative diseases cause a gradual decline in visual function, early therapeutic intervention can significantly improve the quality of life of the patients. There is an urgent need to develop a less invasive treatment that can be implemented for patients with retinal degenerative diseases who have maintained visual function.

In the central nervous system, the administration of small molecule compounds enables the reprogramming of fibroblasts into neurons and neural progenitor cells [9–11]. Mahato et al. showed that fibroblasts could differentiate into retinal progenitor cells [12]. Zhang et al. confirmed that endogenous astrocytes differentiate into neurons in mice whose ventricles were injected with inhibitors [13]. These reports served as motivation to explore the possibility of the differentiation of retinal component cells (i.e., endogenous cells) into photoreceptor cells following the administration of compounds. Intravitreal injection is a less invasive treatment than subretinal injection. In addition, small molecule drugs are less expensive than cell transplantation therapy or gene therapy.

Retinal glial cells are composed of Müller cells (MCs), astrocytes, and microglia [14]. In zebrafish, MCs are a source of new photoreceptor cells after retinal injury [15]. Unfortunately, mammalian MCs do not possess the ability to generate new photoreceptor cells. However, Ueki et al. [16] and Jorstad et al. [17] reported that MCs could be reprogrammed into retinal neurons in mice via gene transfer of *Ascl1*; this gene is essential for the differentiation from glial cells to photoreceptor cells in zebrafish, similar to *high mobility group A1* (*HMGA1*) [18]. Additionally, Yao et al. showed MC-derived rod cell production via gene transfer of *β-catenin*, *Otx2*, *Crx*, and *Nrl* [19]. Based on the findings of previous research, we hypothesized that it might be possible to produce MC-derived photoreceptors via stimulation with small molecule compounds.

## Materials and methods

### Animals

Male and female B6 wild-type (WT) mice, B6.Cg-Gt(ROSA)26Sortm14(CAG-tdTomato)Hze/J (ROSA-td-Tomato) mice, and B6.CXB1-Pde6βrd10/J (rd10) mice were used. ROSA-td-Tomato and rd10 mice were purchased from the Jackson Laboratory. The mice were bred and reared under cyclic light (12 hours light: 12 hours dark). To perform the experiments, mice were anesthetized via intraperitoneal injection of 15 mg/kg ketamine and 7 mg/kg xylazine. The chemical-induced retinal degeneration model was created via intraperitoneal administration of a single dose of N-methyl-N-nitrosourea (MNU) to mice. After anesthesia, the mice were weighed, and injected with MNU (30–75 mg/kg). All mice were treated in accordance with the standards of the Association for Research in Vision and Ophthalmology for the use of animals in ophthalmic and vision research. All animal experiments were reviewed and approved by the Kyushu University Ethics Committee for Animal Experimentation and were conducted in accordance with the relevant guidelines and regulations (A20-360, A21-225).

## Cell culture

Primary retinal MCs were cultured as previously described [20]. Briefly, the eyes of 6-week-old WT mice were enucleated under sterile conditions. The isolated retinas were then digested using collagenase D (1.2 mg/mL, Roche, Mannheim, Germany) and DNAse I (0.005%, Roche). Cells were seeded on a poly L-Lysine-coated dish (Corning, NY, USA) and cultured at 34˚C and 5% $CO_2$. The culture medium was composed of Dulbecco's Modified Eagle Medium (DMEM) high glucose (Sigma-Aldrich, St Louis, MO, USA), 1% penicillin-streptomycin, and 10% fetal bovine serum. The rat MC cell line (TR-MUL5) was purchased from Fact, Inc. (Sendai, Japan) [21] and cultured in the same culture medium.

## *In vitro* stimulation of MCs and TR-MUL5 cells with small molecule compounds

One day before the administration of the compounds, half of the culture medium was replaced with the stimulating medium composed of Ham's F-12 (FUJIFILM Wako, Osaka, Japan), N2 supplement (Thermo Fisher Scientific, Waltham, MA, USA), and 1% penicillin-streptomycin. The next day, all of the medium was replaced with stimulating medium. Five compounds, namely, SB431542 (18176–54, 5 μM, Nacalai, Kyoto, Japan), LDN193189 (6053–10, 1 μM, R&D Systems, Minneapolis, MN, USA), CHIR99021 (13122, 1.5 μM, Cayman, Ann Arbor, MI, USA), DAPT (208255-80-5, 5 μM, Sigma-Aldrich), and Y-27632 (18188–04, 0.5 μM, Nacalai), were added to the stimulating medium. The medium was changed every two days and the compounds were added at the same time.

## Intravitreal injection

After anesthesia, the compounds were diluted in phosphate buffered saline (PBS, pH 7.4) and injected into the vitreous of 6-week-old WT, ROSA-td-Tomato, and 2- and 4-week-old rd10 mice under topical mydriasis at 9 AM. The final concentrations of SB431542, LDN193189, CHIR99021, and DAPT were 0.1, 0.02, 0.03, and 0.1 μM, respectively. Micro syringes with 33-gauge needles (Hamilton, Reno, NV, USA) were used. After anesthesia, we intravitreally injected 1 μL of pAAV.GFAP.Cre.WPRE.hGH at a concentration of $7 \times 10^{13}$ vg/μL into 2-week-old ROSA-td-Tomato mice. PAAV.GFAP.Cre.WPRE.hGH was a gift from James M. Wilson (Addgene plasmid # 105550; http://n2t.net/addgene:105550; RRID:Addgene_105550).

## Quantitative real-time polymerase chain reaction (qPCR)

The total RNA of the retinal tissue and cultured cells was extracted using the NucleoSpin RNA kit (Macherey-Nagel, Düren, Germany) according to the manufacturer's instructions. RNA was reverse transcribed using the First Strand cDNA Synthesis Kit for RT-PCR (Roche). Real-time qPCR was performed using SYBR Premix Ex Taq (Takara, Shiga, Japan) and a LightCycler 96 (Roche). The primer sequences used are listed in S1 Table. Retinal neuron types and their corresponding markers are summarized in S2 Table [22–28].

## Immunocytochemistry

Cultured cells were fixed using 4% paraformaldehyde (PFA) at room temperature for 20 min. After washing with PBS, cells were incubated with 0.3% Triton X-100 (FUJIFILM Wako) in PBS for 15 min and blocked with PBS containing 10% normal goat serum (NGS) (Thermo Fisher Scientific) for 1 h at room temperature. Cells were then incubated with primary antibodies (chicken anti-vimentin [polyclonal, 1/1000, ab24525, Abcam, Cambridge, UK], rabbit anti-glutamine synthetase [GS] [polyclonal, 1/100, ab73593, Abcam], mouse anti-rhodopsin

[Rho] [monoclonal, 1/100, ab5417, Abcam], and rat anti-mouse CD44-PE [monoclonal, 1/100, 1M7, eBioscience, San Diego, CA, USA]) diluted in PBS at 4˚C overnight. Next, the cells were incubated with Alexa-conjugated secondary antibodies (Goat anti-Chicken IgY, [H+L], Alexa Fluor™ 488, A-11039, Goat anti-Rabbit IgG [H+L], Alexa Fluor™ 546, A-11035, Goat anti-Mouse IgG [H+L], Alexa Fluor™ 647, A-21235, Thermo Fisher Scientific) diluted in PBS for 1 h at room temperature, and the nuclei were counterstained with 4′,6-diamidino-2-pheny-lindole (DAPI). Immunofluorescence images were acquired using a BZ-X710 confocal micro-scope (Keyence, Osaka, Japan). For cell counting, at least three fields were selected per dish.

## Immunohistochemistry (frozen section)

The eyes were enucleated, fixed with 4% PFA for 30 min at room temperature, and immersed in OCT compound. Then, they were frozen and cut into 8-μm sections using a cryostat (CM1800, Leica Microsystems, Wetzlar, Germany) and mounted onto MAS-coated glass slides (MAS-01, Matsunami, Osaka, Japan). After removing the OCT compound in PBS containing 0.3% Triton X-100 for 15 min, the sections were blocked with 10% NGS in PBS for 1 h. Then, they were incubated with primary antibodies (chicken anti-vimentin [ab24525, Abcam], mouse anti-Rho [1D4, Abcam], FITC-conjugated peanut agglutinin (PNA) [L7381, Sigma-Aldrich], rabbit anti-PSD95 [monoclonal, 1/100, D27E11, Cell Signaling Technology, Danvers, MA, USA], rabbit anti-GS [ab73593, Abcam]) diluted in PBS at 4˚C overnight. After washing with PBS, the sections were incubated with Alexa-conjugated secondary antibody (Goat anti-Rabbit IgG [H+L], Alexa Fluor™ 488, A-11008, Goat anti-Rabbit IgG [H+L], Alexa Fluor™ 546, A-11035, Goat anti-Mouse IgG [H+L], Alexa Fluor™ 647, A-21235) diluted in PBS for 1 h at room temperature, and the nuclei were counterstained with DAPI. Images were acquired using a BZ-X710 confocal microscope.

## Immunohistochemistry (paraffin sections)

After fixation with 4% PFA for 30 min at room temperature, the eyes of mice were embedded in paraffin and sliced into 3-μm sections. After removing the paraffin, the sections were acti-vated, blocked, and incubated with primary antibodies (chicken anti-vimentin [ab24525, Abcam], mouse anti-Rho [1D4, Abcam], FITC-conjugated PNA [L7381, Sigma-Aldrich], rab-bit anti-PSD95 [D27E11, Cell Signaling Technology, rabbit anti-GS [ab73593, Abcam]) diluted in PBS at 4˚C overnight. After washing with PBS, the sections were incubated with Alexa-con-jugated secondary antibody (Goat anti-Chicken IgY, [H+L], Alexa Fluor™ 488, A-11039, Goat anti-Rabbit IgG [H+L], Alexa Fluor™ 488, A-11008, Goat anti-Rabbit IgG [H+L], Alexa Fluor™ 546, A-11035, Goat anti-Mouse IgG [H+L], Alexa Fluor™ 647, A-21235) diluted in PBS for 1 h at room temperature, and the nuclei were counterstained with DAPI. Images were acquired using a BZ-X710 confocal microscope. The Rho-positive length in the outer retinal layers was analyzed using ImageJ software. At least three sections, including the optic nerve, were mea-sured per eye, and the average was calculated.

## Fluorescent activated cell sorting (FACS)

For the flow cytometry measurements, cultured cells were washed with PBS containing 0.25% trypsin, and the cells were incubated for 3 min at 37˚C. After washing cells with 2% bovine serum albumin (BSA) in PBS, the cells were fixed with 1% PFA and permeabilized in 0.3% Tri-ton X-100 for 10 min at 4˚C. After rinsing, the cells were incubated with antibodies (rat anti-vimentin-APC [IC2105A, R&D Systems], rat anti-CD44-PE [eBioscience]) for 1 h at 4˚C. After rinsing, the cells were resuspended in 2% BSA in PBS and analyzed using FACS. FACS-Verse and FlowJo software v10 (BD Biosciences, Franklin Lakes, NJ, USA) were used for the

analysis. For td-Tomato-positive cell sorting, the retinas isolated from ROSA-td-Tomato mice were incubated with collagenase D (1.2 mg/mL) and DNAse I (0.005%) for 20 min at 37°C. After washing with 2% BSA in PBS, the cells were stained with a 7-Amino-Actinomycin D Viability Staining Solution (BioLegend, San Diego, CA, USA) to remove the dead cells. A BD FACSAria™ III cell sorter (BD Biosciences) was used for live td-Tomato-positive cell sorting.

## TUNEL staining

TUNEL staining was performed using an ApopTag fluorescein *in situ* apoptosis detection kit (Merck Millipore, Darmstadt, Germany) according to the manufacturer's instructions. The number of TUNEL-positive cells in the outer nuclear layer (ONL) was counted as described for MNU-treated mice [29]. Briefly, the TUNEL-positive cells in the ONL of a 9,000 (150 × 60) $\mu m^2$ area, 400 μm away from the optic nerve papilla and ciliary body, were counted in a masked fashion. Then, we calculated the average number of TUNEL-positive cells based on that of four fields for each section.

## 5-bromo-2′-deoxyuridine (BrdU) staining

BrdU staining of MCs was performed in accordance with a previous report [30]. Mice were administered BrdU (50 μg/g BW, ab142567, Abcam) via intraperitoneal injection every day for 7 days. After treatment with BrdU, the eyes were enucleated, fixed with 4% PFA for 30 min at room temperature, and then frozen sections were prepared. The sections were pretreated with 2 N HCl for 30 min at 37°C before being blocked with 10% NGS in PBS for 1 h. Sections were then incubated with rat anti-BrdU (ab6326, Abcam) at 4°C overnight. After washing with PBS, the sections were incubated with an Alexa-conjugated secondary antibody for 1 h at room temperature, and the nuclei were counterstained with DAPI. Images were acquired using a BZ-X710 confocal microscope.

## Subretinal transplantation of MCs

Td-Tomato-positive MCs were collected from ROSA-td-Tomato mice and cultured [20]. After anesthesia, 1 μL of PBS with or without $2 \times 10^5$ td-Tomato-positive MCs was injected transvitreally into the subretinal space of 6-week-old WT mice.

## Retinal whole-mount staining

Eyes were enucleated and fixed with 4% paraformaldehyde for 1 h at 4°C. The isolated retinas were incubated with PBS containing 0.5% Triton X-100 for 30 min and blocked for 1 h with PBS containing 10% NGS and 0.5% Triton X-100. The retinas were then incubated with FITC-conjugated PNA (L7381, Sigma-Aldrich) at 4°C overnight. After washing with PBS, the retinas were mounted on slides. To assess the number of cone photoreceptor cells, we counted the PNA-positive cells in $125 \times 125$ $\mu m^2$ retinal areas in the superior, inferior, temporal, and nasal areas located 250 μm, 500 μm, and 1,000 μm away from the optic disc. The names and conditions of the samples were masked to the observers. The number of PNA-positive cells was averaged from those in 12 retinal areas for each retina. Immunofluorescence images were acquired using a BZ-X710 confocal microscope.

## Electroretinography (ERG)

Electroretinograms (ERGs) were recorded as described previously [31]. Briefly, mice were dark-adapted overnight and anesthetized under dim red light. After mydriasis, recordings were made using a PuREC system (Mayo, Aichi, Japan), as previously reported [32]. ERGs

were recorded on day 7 after intravitreal injections of the four compounds, according to the International Society for Clinical Electrophysiology of Vision standard protocol by using five light stimuli per recording. The scotopic ERGs were elicited using a stimulus intensity of 10,000 cd/m$^2$. The responses were differentially amplified and filtered in the 0.3 to 500 Hz range.

### Statistical analysis

Data are presented as the mean ± standard error of mean (SEM). Statistical analysis was performed using JMP Pro version 16.0 (SAS Institute). The significance of the difference was determined using Wilcoxon rank sum test. Differences were considered statistically significant at p < 0.05. The number of biological repeats (n) for each experiment is presented in the figure legends.

## Results

### Screening of the small molecule compounds used for converting MCs into photoreceptor cells

We isolated the MCs from 6-week-old B6J mice, according to previous reports (S1 Fig) [20]. We examined the changes in the expression of photoreceptor markers and transcription factors in primary MCs after stimulation with candidate compounds to test whether the compounds could chemically differentiate MCs into cells that express photoreceptor-specific mRNAs or proteins.

We selected five compounds that act on signaling pathways that are important for neural differentiation; these have been used in multiple studies [9, 10, 13, 33–37]. The selection is based on previous reports on the successful differentiation of somatic cells, including fibroblasts and astrocytes, into neurons by using small molecule compounds. The compounds used were the following: SB431542 (transforming growth factor-β [TGF-β] inhibitor), LDN193189 (bone morphogenetic protein [BMP] inhibitor), CHIR99021 (glycogen synthase kinase-3β [GSK-3β] inhibitor), DAPT (γ-secretase inhibitor), and Y-27632 (ROCK inhibitor).

The time course of the *in vitro* experiments is shown in Fig 1A. First, the expression of photoreceptor markers was examined using PCR analysis 7 days after the stimulation with all 31 combinations of the five compounds (Fig 1B). The expression level of the messenger RNA (mRNA) of short-wavelength opsin (cone cell marker) was below the detection sensitivity for all compound combinations. On the other hand, the expression level of Rho (rod cell marker) mRNA was markedly increased to about 30-fold of the control in the combination of SB431542 (S), LDN193189 (L), CHIR99021 (C), and DAPT (D) (Fig 1B). The immunocytochemistry results showed that these four compounds (SLCD treatment) induced the expression of Rho protein in 25% of living cells (Fig 1C and 1D). The expression levels of the genes that are regulated by these inhibitors were also evaluated. The SLCD treatment increased the expression of Axin2 (a downstream molecule of the Wnt/ β-catenin pathway) and decreased the expression of LTBP1 (a downstream molecule of the TGF-β pathway), Id (a downstream molecule of the BMP pathway), and DLL1 (a downstream molecule of the Notch pathway), suggesting that the signaling pathways corresponding to each compound were inhibited (S2 Fig).

The morphology of the MCs gradually changed from day 4, and a neurite-like structure was observed on day 7 (Fig 1E). The expression of CD44, an MC marker [38], significantly decreased with the change in cell morphology (Fig 1F). Ascl1 and HMGA1, which are essential for differentiation to photoreceptor cells in zebrafish [16, 18], were significantly upregulated after stimulation (S3 Fig). We also confirmed that the expression of SOX2, a retinal progenitor

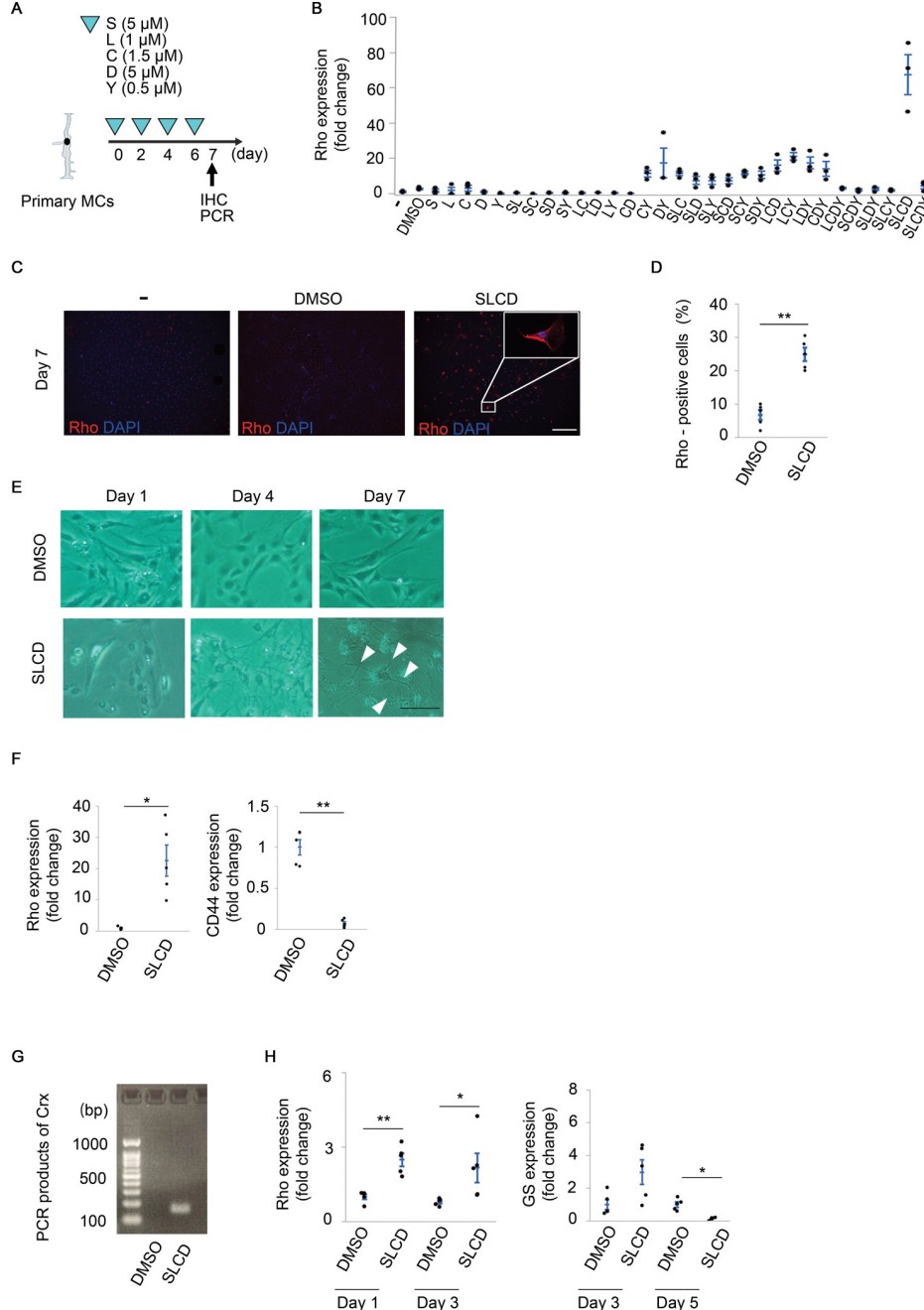

**Fig 1. Screening of small molecule compounds for converting primary MCs into photoreceptor-like cells.** A) Time course of the *in vitro* experiments. MCs were stimulated with five compounds every 2 days. S, SB431542 (5 μM); L, LDN193189 (1 μM); C, CHIR99021 (1.5 μM); D, DAPT (5 μM); Y, Y-27632 (0.5 μM). B) Comparison of the Rho mRNA expression levels for all 31 combinations using real-time qPCR (n = 3). C) and D) Immunofluorescence of Rho (C) and the corresponding quantitative results (D) on day 7 after the stimulation with SLCD. Scale bar = 100 μm (n = 5). E) Microscopic images of the cultured cells. A neurite-like structure (arrowheads) could be observed on day 7 after the stimulation with SLCD (n = 5). Scale bar = 100 μm. F) Real-time qPCR results of Rho and CD44 on day 7 after the stimulation with SLCD (n = 5). G) PCR products of Crx on day 7 after SLCD stimulation. The Crx expression level was below the detection sensitivity in the control group (n = 5). H) Real-time qPCR results of the Rho and glutamine synthetase (GS) expression in TR-MUL5 (n = 5). All data are presented as the mean ± SEM. *p < 0.05, **p < 0.01, as per Wilcoxon rank sum test.

marker [39], increased 1 day after stimulation with SLCD (S3 Fig). In addition, we quantified the level of each specific marker of retinal neurons, other than photoreceptors. As shown in Fig 1F, 1G and S4 Fig, only the expression of Rho and Crx was upregulated. The phenomenon of an increased Rho expression and a decreased expression of glial cell marker was observed not only in primary mouse cells, but also in rat cell lines (Fig 1H). These results suggest that MCs can be differentiated into rod-like cells by the simultaneous administration of SLCD.

## Induction of Rho expression in the outer retina and restoration of retinal function in MNU-treated mice

First, we used a mouse model of retinal degeneration chemically induced by MNU, that is a potent carcinogen, teratogen, and mutagen [40]. MNU damages the outer retinal layers, such as the retinal pigmented epithelium (RPE) layer, the ONL, and OPL, while causing little damage to the inner retinal layers. MNU induces progressive photoreceptor apoptosis, RPE degeneration, and subretinal fibrosis; it has been widely used in research to elucidate the pathogenesis of RP and AMD [41].

We determined the dose of MNU at which the ONL disappeared 7 days after intraperitoneal injection (S5 Fig). We then performed intravitreal injections of SLCD after the administration of 75 mg/kg MNU to investigate whether photoreceptor cells were observed in the outer retinal layer (S5 Fig). We first tested for the appearance of Rho-positive cells by intravitreal injection of SLCD on days 7, 10, and 13 after MNU administration (S6 Fig). We observed the appearance of Rho-positive cells between the ONL and RPE, but could not confirm the recovery of retinal function by full-filed ERG (S6 Fig). We aim to develop an early therapeutic agent that can intervene at a different stage than cell transplantation. Therefore, we decided to determine whether the recovery of retinal function can be achieved with the appearance of Rho-positive cells by SLCD injections on days 0, 3, and 6 after MNU administration (Fig 2A).

The IHC results are shown in Fig 2B. The level of PNA, which selectively binds to the cone inner and outer segments, did not differ between the control and SLCD-stimulated groups; however, the Rho expression was significantly increased in the SLCD-stimulated group (Fig 2B and 2C). The quantification of the percentage of the Rho-expressing retinal region in the total retinal length showed that Rho expression was significantly higher after the intravitreal injection of SLCD (Fig 2D).

Subsequently, we tested whether the difference in the expression of Rho-positive cells between the control and SLCD-stimulated groups depended on creation of photoreceptor cells or inhibition of photoreceptor cell death, particularly apoptosis induced by MNU [34]. The PCR results showed a significant increase in the Rho mRNA level only on day 7 after the administration of SLCD (Fig 2E and S7 Fig) however, the number of apoptotic cells did not change (Fig 2F and 2G). In addition, the cells that were co-positive for BrdU and Rho were observed on day 7 after the administration of SLCD (Fig 2H). These data suggest that the Rho-positive cells may be newly formed.

We used a lineage tracing model that expressed the td-Tomato protein to track MCs, to confirm that the Rho-positive cells originated from the MCs. PAAV.GFAP.Cre.WPRE.hGH, which is an adeno-associated virus (AAV) containing the human GFAP promoter, was injected into the vitreous of 2-week-old ROSA-td-Tomato mice, so that the MCs could be visualized. Four weeks after the injection into the vitreous, the IHC results showed that td-Tomato co-localized with GS, indicating the introduction of AAV into the MCs (S8 Fig). On day 7 after the MNU administration, td-Tomato was co-expressed with Rho in the outer retinal layer only in the SLCD-stimulated groups, suggesting that the Rho-positive cells originated from MCs (Fig 2I). The fluorescent-activated cell sorting (FACS) results showed that the

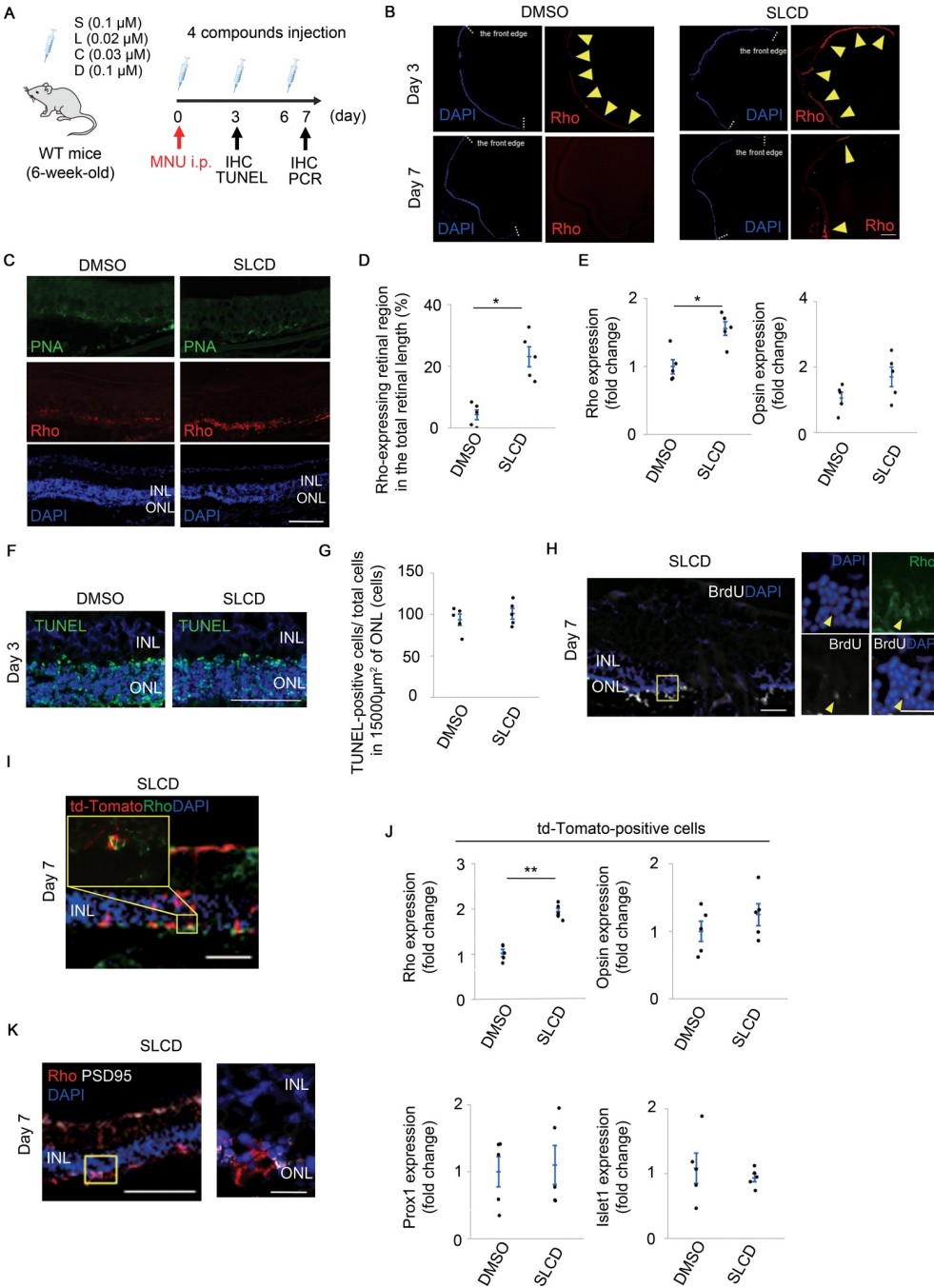

**Fig 2. Differentiation of MCs into Rho-positive cells in MNU-treated mice via intravitreal injection of SLCD.** A)
Time course of the *in vivo* experiments using MNU-treated mice. After a single systemic administration of 75 mg/kg
MNU, intravitreal injection of SLCD was performed every 3 days. i.p., intraperitoneal injection; TUNEL, TdT-
mediated dUTP nick end labeling. B) Immunohistochemistry results of Rho (arrowheads) and DAPI on days 3 and 7
after the intravitreal injection of SLCD into MNU-treated mice. White dotted lines indicate the anterior edge of the
retina. Scale bar = 400 μm. C) Magnified images of Rho (red) and PNA (green) expression in (B) (n = 5). Scale
bar = 100 μm. D) Quantitative results of the Rho-expressing retinal region in the total retinal length (B) (n = 5). E)
Real-time qPCR results of the opsin and Rho expression in the whole retina on day 7 after the intravitreal injection of
SLCD (n = 5). F) and G) TUNEL staining (F) and quantification of the TUNEL-positive photoreceptor cells (G) on day
3 after the intravitreal injection of SLCD (n = 5). Scale bar = 100 μm. H) The Rho-positive cells in the ONL (yellow
arrowhead) also expressed BrdU on day 7 after the intravitreal injection of SLCD. Scale bar = 50 μm. Magnified images
of yellow-framed are shown on the right (n = 5). Scale bar = 20 μm. I) Td-Tomato (red) was co-expressed with Rho
(green) in the outer retinal layer on day 7 after intravitreal injection of SLCD into MNU-treated GFAP-td-Tomato

mice (n = 5). Scale bar = 50 μm. J) Real-time qPCR results of the expression of Rho and other retinal neuron-specific markers in the td-Tomato-positive cell population on day 7 after the intravitreal injection (n = 5). K) PSD95 (white) was also expressed around Rho-positive cells between INL and ONL. Scale bar = 100 μm. A magnified image of yellow-framed is shown on the right (n = 5). Scale bar = 20 μm. OPL, outer plexiform layer. All data are presented as the mean ± SEM. *p < 0.05, **p < 0.01, as per Wilcoxon rank sum test.

SLCD treatment increased the expression level of Rho but did not change the expression levels of other neural markers in td-Tomato-positive cells (Fig 2J). These results were consistent with those obtained for primary cells, suggesting that the td-Tomato-positive MCs were converted into rod-like cells, not into other retinal neurons, after the administration of SLCD. Postsynaptic density protein 95 (PSD95), a synaptic marker, was also expressed around newly produced Rho-positive cells in the OPL of the retina (Fig 2K). We also confirmed that Rho-positive cells remained significant on day 14 even after the SLCD administration was stopped by day 6 (S9 Fig).

Next, we investigated whether MCs have the potential to express Rho after subretinal transplantation. The experimental protocol is shown in Fig 3A. We purified only the td-Tomato-positive MCs from the retinas of WT mice injected intravitreally with AAV, according to a previous report [20]. The td-tomato positive MCs were transplanted under the retina. The compounds were then injected intravitreally on day 4 after MC transplantation. As shown in Fig 3B, the td-tomato positive cells attached to the RPE expressed Rho on day 7 after the SLCD stimulation. These results support the possibility of the differentiation of MCs into rod-like cells by the SLCD stimulation.

In addition, to evaluate whether SLCD could restore retinal function, the ERG results were recorded 7 days after SLCD were injected into MNU-treated mice. In mice treated with 75 mg/kg MNU, SLCD increased the Rho expression, according to the IHC results (Fig 2B); however, the ERG results did not show an obvious improvement (Fig 4A). Since the ERG waveform disappeared after the administration of 75 mg/kg MNU, it was possible that the retinal function might have been so severe that the effect of SLCD could not be detected. Therefore, the MNU dose was reduced to 30 mg/kg and ERG was performed again. As shown in Fig 4A and 4B, the amplitude of the a-waves was significantly improved by the injection of SLCD. These findings suggest that the Rho-positive cells induced by the simultaneous administration of SLCD to the vitreous restored the retinal function in MNU-treated mice.

## Induction of Rho expression in the outer retina and inhibition of cone cell death in rd10 mice

Finally, we tested whether SLCD treatment produced Rho-positive cells in rd10 mice, a mouse model of RP with a missense point mutation in the β-subunit of the rod cyclic guanosine monophosphate of the *phosphodiesterase* gene (*PDE6β*) [42]. Point mutations in *PDE6β* have been detected in patients with autosomal recessive RP [43]. We performed intravitreal injections of SLCD every 3 days in 4-week-old rd10 mice and investigated the Rho expression changes (Fig 5A). We first quantified the expression of retinal neuron markers in the whole retina using PCR. After treatment with SLCD, only the expression of Rho was upregulated, and the expression levels of other markers remained unchanged (Fig 5B and S10 Fig). Moreover, the IHC results showed an obvious increase in Rho expression in the outer retina of the SLCD-stimulated group (Fig 5C and 5D). We quantified the percentage of the Rho-expressing retinal region in the total retinal length; indeed, the Rho expression was significantly increased by the intravitreal injection of SLCD (Fig 5E).

In rd10 mice, rod cell death peaked at 3–4 weeks, followed by cone cell death [44]. Next, we counted the PNA-positive cone cells based on our previous report [42] to verify whether the

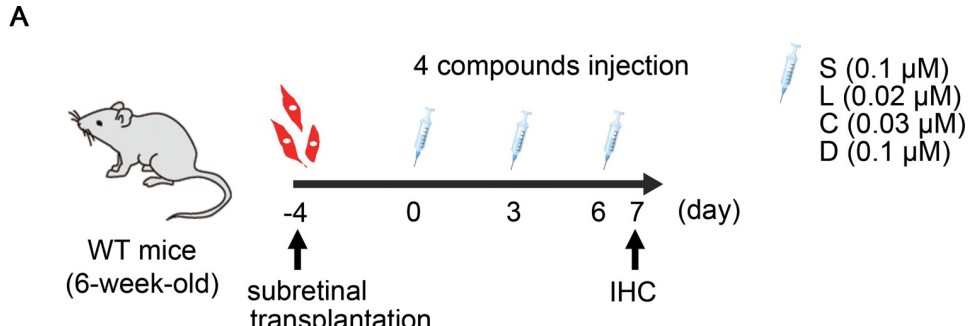

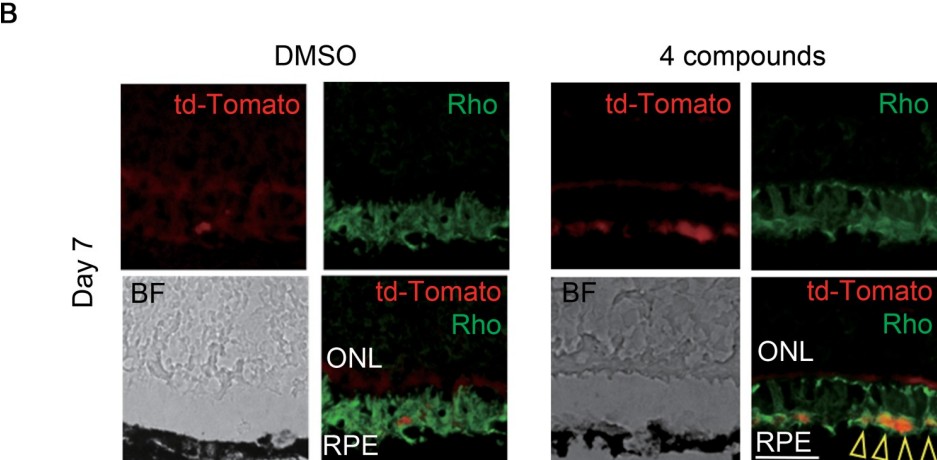

**Fig 3. Rho expression in MCs transplanted into the subretinal space.** A) Time course of the experiments. Td-tomato-positive MCs were transplanted into the subretinal space in 6-week-old WT mice. Injections of SLCD were administered every three days from day 4 after the transplantation. B) Immunohistochemistry of colocalization (yellow arrowheads) of td-Tomato (red) and Rho (green) on day 7 after SLCD stimulation (n = 5). Scale bar = 50 μm.

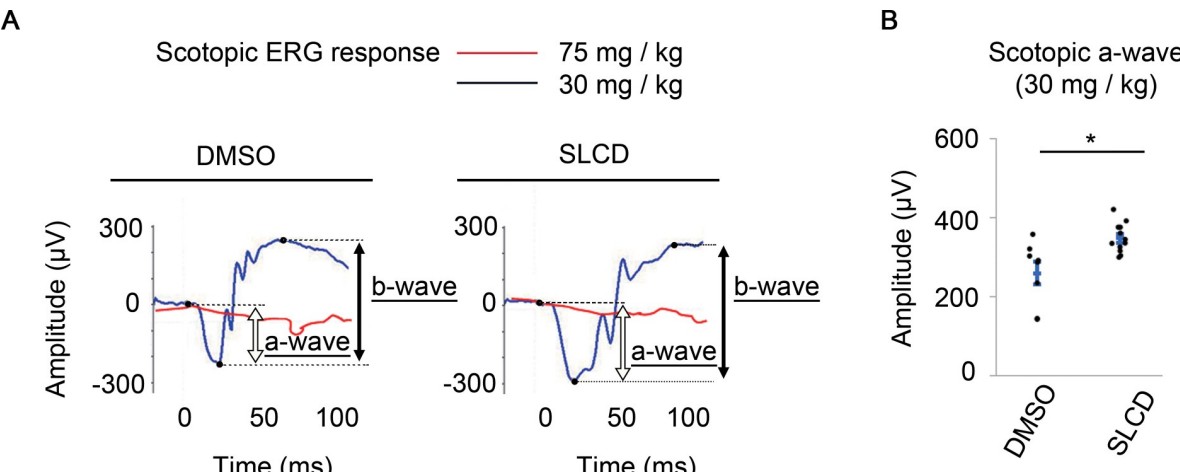

**Fig 4. Improvement of the retinal function by MC-derived Rho-positive cells in MNU-treated mice.** A) Scotopic responses of the ERGs on day 7 after the intravitreal injection of SLCD. ERG waveforms for the 75 mg/kg MNU group are shown in red, and those for the 30 mg/kg MNU group are shown in blue. B) Effect of the compounds on the a-wave in the 30 mg/kg MNU group. All data are presented as the mean ± SEM (DMSO, n = 8; SLCD, n = 12). $^*p < 0.05$, as per Wilcoxon rank sum test.

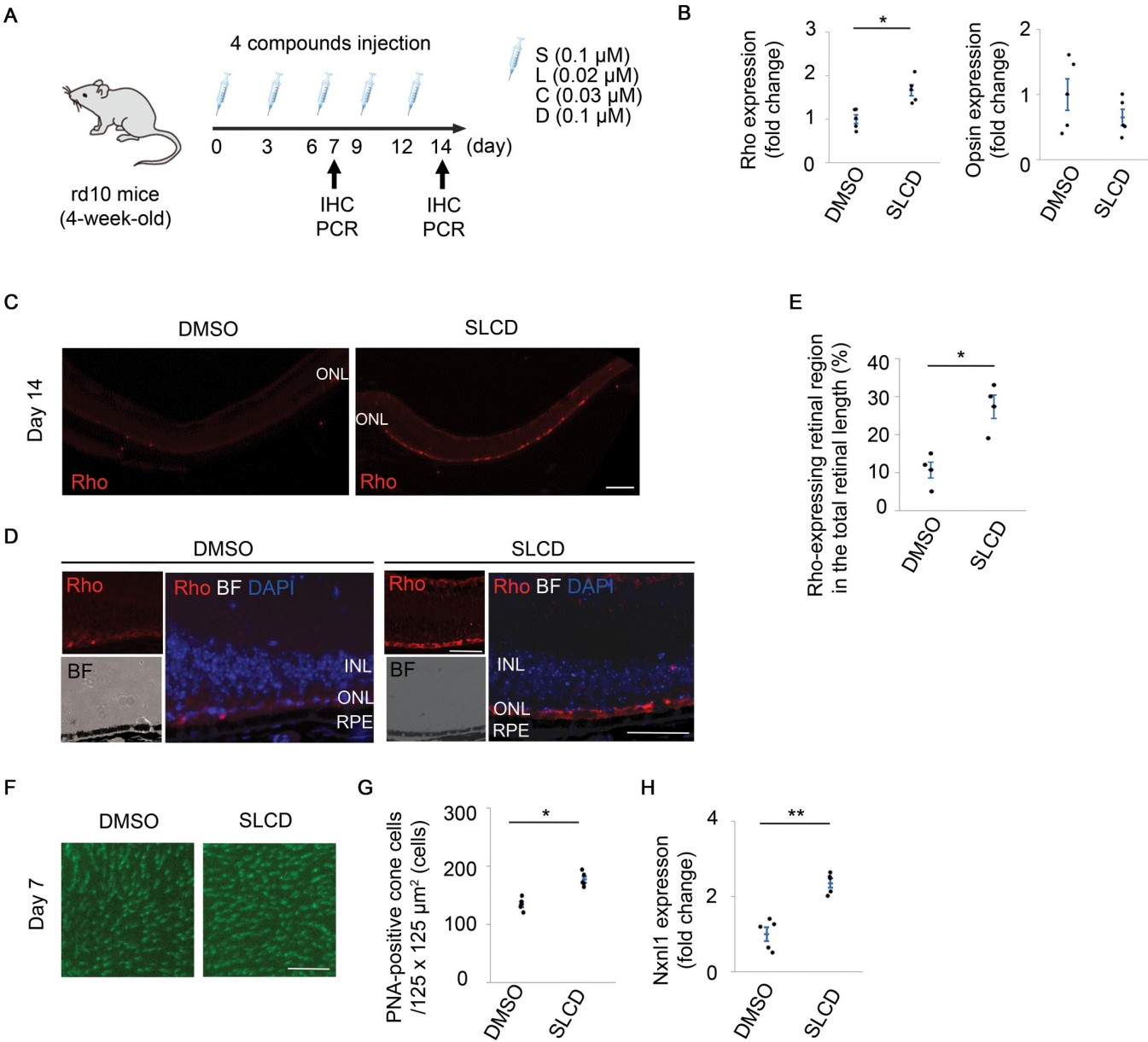

**Fig 5. Contribution of the Rho-positive cells to the suppression of cone cell death in rd10 mice.** A) Time course of the *in vivo* experiments using rd10 mice. An intravitreal injection of SLCD was performed every 3 days for 2 weeks from 4 weeks of age. B) Real-time qPCR results of the opsin and Rho expression in the whole retina on day 7 after the intravitreal injection of SLCD (n = 5). C) and D) Immunohistochemistry results of Rho (C) and quantitative results of the Rho-expressing retinal region in the total retinal length (D) on day 14 after the intravitreal injection of SLCD. Scale bar = 200 μm (n = 5). E) Magnified images of Rho (red) expression and bright field in (C). Scale bar = 100 μm. F) and G) Whole-mount PNA staining (F) and quantification of the PNA-positive cone cells (G) in the outer retinas on day 7 after the intravitreal injection of SLCD (n = 5). Scale bar = 50 μm. H) Real-time qPCR results of NXNL1 on day 7 after the intravitreal injection of SLCD (n = 5). All data are presented as the mean ± SEM. *p < 0.05, as per Wilcoxon rank sum test.

inhibition of cone cell death was associated with an increased expression of Rho-positive cells. Retinal whole-mount staining showed that significantly more PNA-positive cone cells were retained on day 7 after the intraocular injection of SLCD (Fig 5F and 5G). In addition, the expression of nucleoredoxin-like 1 (NXNL1) was significantly increased in the compound-stimulated group (Fig 5H).

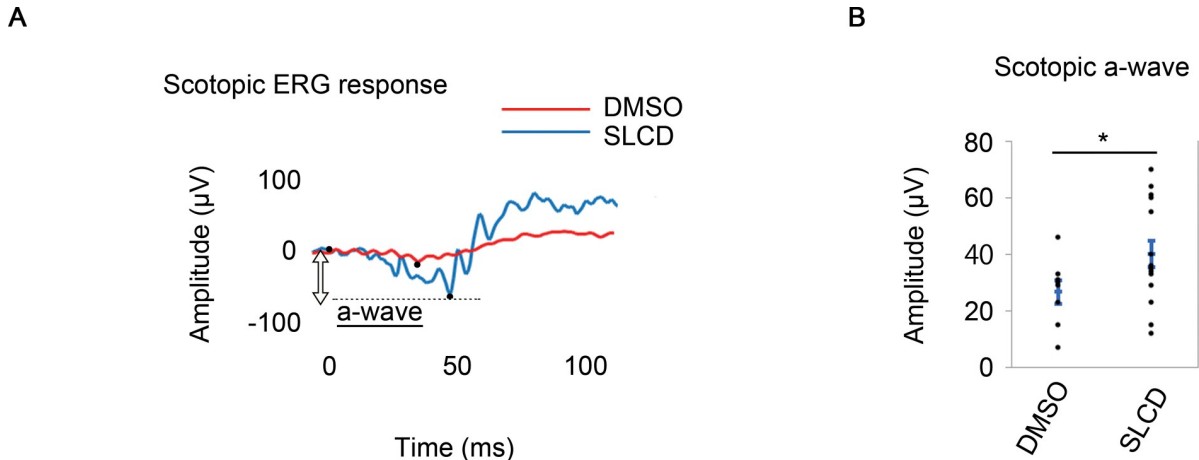

**Fig 6. Improvement of the retinal function by intravitreal injection of SLCD in rd10 mice.** A) Scotopic response of the ERGs on day 7 after the intravitreal injection of SLCD in rd10 mice. ERG waveform for the SLCD-stimulated group is shown in blue, and that for the control group is shown in red. B) Quantitative results of the a-wave in A). All data are presented as the mean ± SEM (DMSO, n = 8; SLCD, n = 14). *p < 0.05, as per Wilcoxon rank sum test.

In rd10 mice, we also investigated whether SLCD administration restored retinal function by ERG. The ERG amplitudes were significantly improved (Fig 6), indicating that visual function improved with the appearance of Rho-positive cells and inhibition of cone cell loss.

## Discussion

In this study, we were able to transform MCs into Rho-positive cells both *in vitro* and *in vivo*. Along with the generation of Rho-positive cells, the amplitude of ERG was restored, suggesting that the Rho-positive cells had some properties similar to those of rod cells. It should be emphasized that endogenous MCs were only differentiated into photoreceptor-like cells through the simultaneous administration of SLCD.

Unlike in the central nervous system, there are no reports of neural differentiation in the mammalian retina solely by direct reprogramming using compounds. However, the signaling pathways that are modified by the compounds used in this study have also been shown to be important in transformation from MCs to retinal neural cells. Osakada et al. demonstrated that the activation of the Wnt/β-catenin signaling pathway by GSK-3β inhibitors promotes the proliferation of MC-derived retinal progenitors after injury [45]. Lenkowski et al. [46] and Todd et al. [47] reported that the TGFb-Smad2/3 signaling pathway regulates production of retinal neurons after injury in zebrafish. They showed that the inhibition of this pathway suppresses MC-derived gliosis and induces the proliferation of MC-derived progenitor cells. In addition, Ueki et al. showed that the activated BMP-Smad1/5/8 signaling pathway promotes MC-derived gliosis in proliferative vitreoretinopathy [48]. Moreover, DAPT increased the expression of Crx, which is important for rod photoreceptor maturation and promotes photoreceptor differentiation.

We found that MCs can be transformed into Rho-positive cells *in vitro* after the elevation of the levels of SOX2, a retinal progenitor marker, and Ascl1 and HMGA1, which are essential transcription factors for neural differentiation in the retina, by the stimulation with SLCD. MC-driven Rho-positive cells did not show an elevated expression of other retinal neural markers, suggesting that MCs could only be induced to transform into rod-like cells. In zebrafish, MC-derived progenitor cells have the ability to differentiate into various types of neurons;

however, they generate unneeded neurons because of their multipotency [49]. Ueki et al. also reported that MCs overexpressing Ascl1 differentiate not only into photoreceptor cells, but also into amacrine and bipolar cells [16]. If SLCD can be administered to induce the differentiation of MCs only into rod-like cells, this therapy will have an advantage over other therapies in terms of safety. MCs play an important role in supporting the function, metabolism, and structure of neurons in the retina. In other words, if MCs are depleted by inducing the differentiation of MCs into nerves, retinal function abnormalities may occur. However, there was no obvious abnormality in the retinal layer structure or loss of function *in vivo* due to compound administration. MCs are capable of asymmetric self-renewal after injury [50, 51], and BrdU experiments have indeed shown that MCs divide and produce Rho-positive cells.

In this study, we chose the "simultaneous" administration of the compounds because we wanted to develop a treatment that was as simple as possible from the point of view of the ophthalmologist. We confirmed that each targeted signaling pathway was properly inhibited, even when SLCD were administered simultaneously. The number of Rho-positive cells decreased in the absence of any of these four compounds and with the addition of an extra compound. These results suggest that the suppression of the TGF, BMP, and Notch signaling pathways and the activation of the Wnt/ β-catenin signaling pathway may have created an environment suitable for the generation of Rho-positive cells. Inhibition of ROCK improves stem cell survival [52], promotes axon elongation [53], and protects MCs from oxidative stress and hypoxia [54]. The reason why SLCD "Y" was less effective than SLCD in inducing Rho expression in MCs is unclear, but competition for inhibitory signals may have prompted a reduction in Rho expression induction. In the present study, the efficiency of the differentiation of primary MCs into Rho-positive cells was only 25%. Single-cell analysis [55, 56] has shown that the MCs were diverse, and it is possible that some MC types are more likely to differentiate into Rho-positive cells upon stimulation with the compounds. In addition, in most previous reports, when multiple compounds were used to increase the efficiency of the differentiation of non-neuronal cells into neuronal cells, the timing of administration of each compound differed. When multiple compounds were administered simultaneously, as in the present study, it may have been difficult to strictly control the resulting signals.

In *in vivo* experiments, even after the rod cells were almost eliminated by MNU, the appearance of Rho-positive cells was observed after intravitreal administration of SLCD. Unfortunately, compound administration did not restore retinal function, suggesting that this therapy is less effective for patients with the retinal degenerative disease who have significantly reduced visual function. Cell transplantation therapy or gene therapy may be more effective for such late stage patients. However, these therapies are highly invasive and difficult to implement in patients with preserved visual function. Therefore, we investigated the effects of these four compounds in the stage of progressive photoreceptor cell death using MNU-treated and rd10 mice.

In MNU mice, the amplitude of the ERG was recovered with the appearance of Rho-positive cells. Sánchez-Cruz et al. previously reported the neuroprotective effect of the GSK-3β inhibitor in 2-week-old rd10 mice [57]. The possibility that photoreceptor protection in the present study contributed to ERG recovery in MNU mice cannot be ruled out; however, unlike in previous reports, a sufficient MNU amount was administered to completely eliminate the ONL, and the number of apoptotic cells in the outer layer did not change between the control and compound-stimulated groups. PSD-95 was expressed around the Rho-positive cells produced from the MCs, suggesting that the new production of photoreceptor-like cells by the SLCD stimulation affected the ERG amplitude recovery.

In rd10 mice, the ERG amplitude also improved, accompanied by the appearance of Rho-positive cells and retention of cone cell number. It was also possible to suppress the apoptosis

of cone cells in rd10 mice, which reveals the potential clinical application of the intravitreal injection of these compounds. Rod cells are sensitive enough to detect a single photon and are suitable for night vision. However, these rarely function in bright areas because they are saturated by moderately bright light. On the other hand, cone cells are 100 times less sensitive to light than rod cells; thus, they do not function in the dark. However, cone cells have an excellent ability to adjust their sensitivity; therefore, they do not saturate, even in bright areas. Generally, visual acuity and color vision depend on the function of the cone cells [44]. It is known that rod cell death is followed by cone cell death in RP; however, the cause of cone cell death is not clear [58]. Léveillard et al. and Mohand-Said et al. reported that a decrease in the secretion of survival factors from rod cells is the cause of cone cell death [59, 60]. Byrne et al. have shown that cone cell death can be suppressed by gene transfer of the rod-derived cone viability factor (RdCVF), which is encoded by *NXNL1* [61]. This study suggests that the secretion of RdCVF from MC-derived Rho-positive cells may inhibit the death of cone cells. If the cone cells of patients with RP can be maintained, the quality of vision will be improved. The rate of degeneration is slower in human RP than in rd10 mice, allowing more time for therapeutic intervention, enabling the treatment of more patients. Since all somatic cells in rd10 mice have a point mutation in *PDE6β*, it was possible that the Rho-positive cells that differentiated from MCs could also undergo apoptosis. Oxidative stress after rod cell death is a commonly accepted cause of cone cell death [58], and there was the risk that compound administration could promote cone cell death. In the present study, however, the SLCD stimulation maintained Rho expression and inhibited cone cell death, possibly because the Rho-positive cells differentiated from MCs are "rod-like" cells and may not die in the same way as native rod cells.

The limitations of this study are as follows. First, the MNU-treated mice and rd10 mice do not fully reflect human RP and AMD. The MNU treatment results in the destruction of the outer retinal layer in approximately one week. In other words, there is a difference between the phenomena that occur in the retinas of MNU-treated mice and those of patients with slowly progressing RP and AMD. Although rd10 mice have the same mutation as humans, the same results may not be obtained for all types of RP. Second, cell tracing experiments cannot precisely follow the origin of Rho-positive cells. Although GFAP-expressing cells were traced using AAV, it is not possible to determine exactly whether the Rho-positive cells originated from MCs or astrocytes. However, the results of the *in vitro* and subretinal transplantation experiments indicate that at least MCs have the potential to express Rho. Third, the possibility of retinal restoration due to the protection of residual photoreceptor cells cannot be completely ruled out. Inhibition of rod cell apoptosis by stimulating Wnt signaling in rd10 mice is possible [54, 62]. Since CHIR99021 can activate the Wnt signaling pathway, the recovery of ERG amplitude may have reflected the inhibition of apoptosis. However, since SLCD did not affect the amount of apoptosis and PSD was expressed around Rho-positive cells in MNU-treated mice, the newly generated Rho-positive cells may contribute to the recovery of a-waves. In addition, once-every-three-day intravitreal injections cannot be used in clinical practice. Appropriate dosages and intervals must be established. The fact that the appearance of Rho-positive cells could be maintained after interruption of compound stimulation suggests that multiple treatments over a short period may provide some long-term improvement in visual function.

In this study, we showed that the simultaneous administration of the four compounds alone can differentiate retinal MCs into Rho-positive cells and restore the function of the injured retina. This method can be applied not only to RP and AMD, but also to the photoreceptor degeneration caused by various retinal diseases. The vitreous injection of compounds is expected to be a new therapeutic strategy that does not rely on cell transplantation or gene transfer. Unlike cell transplantation and gene therapy, the intravitreal injection of compounds is inexpensive and easy to re-perform, and it may be applicable to patients at various stages of the disease.

## Supporting information

**S1 Fig. Isolation of the primary MCs from the retina.** A) Purification of the primary MCs. B) Microscopic images of the cultured cells. Scale bar = 200 μm. C) Immunofluorescence of Vim (green) and GS (red) expression on day 24 (n = 5). Scale bar = 100μm. D) Immunofluorescence of Vim (green) and CD44 (red) (n = 5). Scale bar = 100 μm. E) Results of the FACS analysis. Vim+ CD44+ cells accounted for 95.4% of the total cells.
(TIF)

**S2 Fig. The changes in the expression of downstream molecules of each compound.** Real-time qPCR results of downstream molecules in each pathway of SLCD on day 7 after the stimulation. The LTBP1 and DLL1 expression level was below the detection sensitivity in the compound group. The Inhibitor of DNA binding (Id) expression level was significantly decreased, and the Axin2 expression level was clearly increased in the compound group. Axin2, downstream molecule of Wnt/β-catenin pathway; LTBP1, downstream molecule of TGF-β pathway; Id, downstream molecule of BMP pathway; DLL1, downstream molecule of the Notch pathway (n = 5). All data are presented as the mean ± SEM. *p < 0.05, as per Wilcoxon rank sum test.
(TIF)

**S3 Fig. Upregulation of the neural differentiation-related genes after the administration of SLCD.** Real-time qPCR results of the Ascl1, HMGA, and SOX2 expression on day 1 and 4 after the administration of SLCD (n = 5). All data are presented as the mean ± SEM. *p < 0.05, **p < 0.01, as per Wilcoxon rank sum test.
(TIF)

**S4 Fig. Effect of SLCD on expression of the retinal neural markers in cultured MCs.** Real-time qPCR results of the retinal neuron-specific markers in cultured MCs on day 7 after the administration of SLCD (n = 5). All data are presented as the mean ± SEM.
(TIF)

**S5 Fig. The differences in the outer retinal damage due to the dose of MNU.** Immunohistochemistry of PNA (green) and Rho (red) on day 3 and 7 at 50, 75, and 100 mg/kg MNU-treated mice respectively. It has been shown that 75 mg/kg is sufficient to cause loss of photoreceptors on day 7 post-dose (n = 5). Scale bar = 100 μm.
(TIF)

**S6 Fig. Altered Rho expression in the outer retina by SLCD after photoreceptor cell disappearance.** A) Time course of the experiments. After an administration of 75 mg/kg MNU, an intravitreal injection of SLCD was performed every 3 days from day 7. B) Immunohistochemistry of PNA (green), Rho (red), and DAPI (blue) after intravitreal injection of SLCD on days 7, 10, and 13 after MNU administration. (n = 5). Scale bar = 100 μm.
(TIF)

**S7 Fig. The effects of SLCD on expression of the neural markers, other than Rho in MNU-treated mice.** Real-time qPCR results of the retinal neuron-specific markers in the whole retina on day 3 and 7 after the intravitreal injection of SLCD (n = 5). All data are presented as the mean ± SEM.
(TIF)

**S8 Fig. Introduction of the td-Tomato proteins into MCs by Cre-LoxP system.** A) Time course of the *in vivo* experiments for MC lineage tracing using pAAV.GFAP.Cre.WPRE.hGH and Rosa-td-Tomato mice. Four weeks after the AAV injection, we tested whether td-Tomato could be efficiently induced into MC. AAV, pAAV.GFAP.Cre.WPRE.hGH. B) Schematic

images of the introduction of the td-Tomato protein to track MCs using the Cre-LoxP system. C) Immunohistochemistry of colocalization of td-Tomato (red) and GS (green) in Rosa-td-Tomato mice four weeks after the AAV injection (n = 5). Scale bar = 100 μm.
(TIF)

**S9 Fig. Persistence of Rho expression after discontinuation of SLCD treatment.** A) Time course of the experiments using GFAP-td-Tomato. After an administration of 75 mg/kg MNU, intravitreal injection of SLCD was performed at day 0, 3 and 6. B) Immunohistochemistry of Rho (white) and DAPI (blue) after intravitreal injection of SLCD on days 7, 10, and 13 after MNU administration. (n = 5). Scale bar = 100 μm.
(TIF)

**S10 Fig. Effects of SLCD on expression of the neural markers, other than Rho in rd10 mice.** Real-time qPCR results of the retinal cell-specific markers in the whole retina on day 7 and 14 after the intravitreal injection of SLCD (n = 5). All data are presented as the mean ± SEM.
(TIF)

**S1 Table. Primer sequences.**
(PDF)

**S2 Table. Lists of markers for the identification of retinal cells in qPCR.**
(PDF)

**S1 Data.**
(PDF)

**S2 Data.**
(PDF)

**S3 Data.**
(PDF)

**S4 Data.**
(PDF)

**S5 Data.**
(PDF)

**S6 Data.**
(PDF)

**S7 Data.**
(PDF)

**S8 Data.**
(PDF)

**S9 Data.**
(PDF)

**S10 Data.**
(PDF)

**S11 Data.**
(PDF)

## Acknowledgments

The authors thank Iori Wada, Mitsuhiro Kurata, Masayo Eto, and Fumiyo Morikawa (Kyushu University) for the technical assistance. The authors also thank Editage (www.editage.com) for English language editing.

## Author Contributions

**Conceptualization:** Mitsuru Arima, Yusuke Murakami.

**Data curation:** Yuya Fujii.

**Investigation:** Yuya Fujii, Mitsuru Arima.

**Methodology:** Yuya Fujii, Mitsuru Arima, Yusuke Murakami.

**Project administration:** Mitsuru Arima, Koh-Hei Sonoda.

**Supervision:** Yusuke Murakami, Koh-Hei Sonoda.

**Validation:** Mitsuru Arima, Yusuke Murakami, Koh-Hei Sonoda.

**Writing – original draft:** Yuya Fujii, Mitsuru Arima.

**Writing – review & editing:** Mitsuru Arima.

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
