## [Decision Letter · Decision Letter 0]

20 Sep 2022

PONE-D-22-16150Rhodopsin-positive cell production by intravitreal injection of

small molecule compounds in mouse models of retinal degenerationPLOS ONE

Dear Dr. Arima,

Thank you for submitting your manuscript to PLOS ONE. After careful consideration, we feel that it has merit but does not fully meet PLOS ONE’s publication criteria as it currently stands. Therefore, we invite you to submit a revised version of the manuscript that addresses the points raised during the review process.

We look forward to receiving your revised manuscript.

Kind regards,

Anand Swaroop

Academic Editor

PLOS ONE

Journal Requirements:

2. As part of your revision, please complete and submit a copy of the Full ARRIVE 2.0 Guidelines checklist, a document that aims to improve experimental reporting and reproducibility of animal studies for purposes of post-publication data analysis and reproducibility: https://arriveguidelines.org/sites/arrive/files/Author%20Checklist%20-%20Full.pdf (PDF). Please include your completed checklist as a Supporting Information file. Note that if your paper is accepted for publication, this checklist will be published as part of your article

“The authors thank Iori Wada, Mitsuhiro Kurata, Masayo Eto, and Fumiyo Morikawa (Kyushu University) for the technical assistance. This study was supported by JSPS KAKENHI [grants number JP18H02956 and JP21H03094] given to KH Sonoda. Research funding was acquired from Senju Pharmaceutical Co., Ltd., given to M Arima, Y Murakami, and KH Sonnoda. The funders had no role in the study design, data collection and analysis, decision to publish, or preparation of the manuscript. The authors also thank Editage (www.editage.com) for English language editing.”

“This study was supported by JSPS KAKENHI [grants number JP18H02956 and JP21H03094] given to KH Sonoda. Research funding was acquired from Senju Pharmaceutical Co., Ltd., given to M Arima, Y Murakami, and KH Sonnoda.The funders had no role in study design, data collection and analysis, decision to publish, or preparation of the manuscript.”

 In your cover letter, please note whether your blot/gel image data are in Supporting Information or posted at a public data repository, provide the repository URL if relevant, and provide specific details as to which raw blot/gel images, if any, are not available. Email us at plosone@plos.org if you have any questions

Reviewers' comments:

Reviewer's Responses to Questions

**Comments to the Author**

1. Is the manuscript technically sound, and do the data support the conclusions?

Reviewer #1: Partly

Reviewer #2: Partly

2. Has the statistical analysis been performed appropriately and rigorously? 

Reviewer #1: Yes

Reviewer #2: No

3. Have the authors made all data underlying the findings in their manuscript fully available?

Reviewer #1: No

Reviewer #2: Yes

4. Is the manuscript presented in an intelligible fashion and written in standard English?

Reviewer #1: Yes

Reviewer #2: Yes

5. Review Comments to the Author

Reviewer #1: The regeneration of retinal cells from glial precursors is currently a very active area of research with a number strategies to induce the change in cell identity under investigation. In the present work by Fujii and colleagues, a set of signaling protein small molecule inhibitors, previously shown to induce neuron differentiation from fibroblasts and astrocytes, is identified that stimulates the expression of rhodopsin in two mouse models of retinal degeneration. The authors report that the Rho expression arises from Muller glia that were triggered by the inhibitor combination to differentiate into photoreceptor-like cells. In one experiment, the combination led to a small functional improvement in scotopic (rod-derived) ERG response. I believe this work provides a starting point for further exploration of these and other small molecules to effect glial cell differentiation changes to achieve neuronal cell renewal. However, the work leaves some important questions unanswered and needs to be improved with respect to methodological descriptions to help ensure other investigators are able to successfully reproduce the findings.

Major questions:

1) In the MNU model, it is unclear to me why the authors chose to begin the SLCD treatment at the onset of the retinal damage induction rather than waiting for the damage to be complete and then starting treatment. The way the experiment is currently performed, it is very difficult to completely rule out that the SLCD isn't exerting a preservation effect on the existing photoreceptors rather than inducing the differentiation of new photoreceptors. Although the authors tested for apoptosis, it is possible the SLCD combo could prevent other types of cell death for example. The fact that the combo is ineffective in preserving ERG function when a MNU dose with high photoreceptor toxicity is used but slightly effective when a dose that causes incomplete photoreceptor cell death, also suggests that the SLCD combo might protect some photoreceptors from degeneration.

2) Why didn't the authors perform ERG analysis on their Rd10 mouse model to assess whether there is a functional benefit?

Other questions and concerns:

3) The authors need to specify the vivarium lighting conditions in the methods, particularly since degeneration in the Rd10 model is affected by the level of illumination.

4) Provide the catalog numbers for the five compounds used in the study

5) Intravitreal injection: the authors need to state the time of day when the compounds were injected as well as the vehicle and the solution pH.

6) Catalog numbers and dilution factors should be given for all antibodies used in the study.

7) I believe the "T" in TGF-b most commonly refers to "transforming", not "tumor"

8) Page 10, line 251: which cone opsin transcript was analyzed.

9) In the figures, the authors might consider referring to their combination as "SLCD" instead of "4 compounds" in order to make it clear which combination is being used in the subsequent studies.

10) Can the authors speculate why inclusion of the "Y" compound (ROCK inhibitor) in the treatment combo is so antagonistic to the effects of the SLCD components?

11) Did the authors ever look past Day 7 in their MNU model to see how durable the expression changes are?

12) Retinal sections in the main figures need to be labeled properly with retinal layers marked.

13) In Figure 3, it would be helpful to indicate what the 75 and 30 mg/kg refers to directly on the figure.

Reviewer #2: Experiments were conducted to test whether incubation with a 4-compound cocktail consisting of SB431542, LDN193189, CHIR99021, and DAPT, which has been shown by others to induce brain glial cells to become neuron-like, can induce Muller cells (MCs) to express a rod photoreceptor phenotype (e.g., expression of rhodopsin) in culture and in vivo, and possibly restore function (e.g., restore ERG a-wave amplitudes) in an MNU damage model.

The manuscript is extremely well-written. The hypothesis, rationale, and overall experimental plan is exciting. Depending on outcomes, rigorous testing could have significant and long-lasting impact on the field. The use of a cell culture model and two in vivo retinal degeneration/damage mouse models, of assessing whether 4-compound treatment results in cells that co-label for BrdU and Rho, and the use of a Td-tomato lineage tracing model provide for a solid design. However, execution of individual experiments is flawed to the point that interpretation of data is equivocal at best. In nearly all experiments, the sampling size is inappropriately small (N = 3 or 4). Culture experiments are easily repeated, neither mouse strain is especially difficult to breed, and the experimental manipulations are standard to any RD lab. This lack of replicates resulted in highly variable observations within cohorts and in many cases, uninterpretable outcomes. Sampling sizes need to be increased. Similarly, in several instances it appears that the data would not pass a normality test, yet statistical tests that require such were chosen. Nonparametric tests should be used, or better yet, replicate numbers for nearly all experiments simply need to be increased.

Overall the experimental design and the data do not exclude the possibility that the 4-compound treatment isn't stimulating Rho expression from vestigial rods in the in vivo models. These deficiencies adds to difficulties in data interpretation.

Similarly, the use of the word or concept of "regeneration" is simply not supported by the data, and critically, could not be supported by the existing experimental design. It may be that the 4-compound treatment induces MCs to express a rod-like phenotype, but even that would not be "regeneration" of photoreceptor cells.

The rd10 model should be more-fully tested by assessing effects of 4-compound treatment on retinal function as measured by ERG (as was attempted in the MNU model).

The following are comments to specific datasets ("F" = "figure") and text by line number; note that several of these are of considerable concern:

Line 34 and many other instances - a cocktail of four (4) compounds is stipulated, but the culture experiments apparently used a cocktail with five (5) compounds. Please explain the discrepancy. Also, please ensure that 4 or 5 is correctly stipulated in all instances in text and figures.

Lines 99/100 compared to Line 108 - It appears that MC cultures were tested with a five (5) compound cocktail of SB431542, LDN193189, CHIR99021, DAPT, and Y-27632 but that the in vivo models were tested with a cocktail of just four (4) compounds, SB431542, LDN193189, CHIR99021, and DAPT, but not Y-27632. Please explain why Y-27632 was not included in in vivo treatments. Also, please discuss the implications to interpretation of outcomes that the culture models were not treated with the same cocktail as the in vivo models.

F1 - The small sampling sizes and very limited marker choices do not allow for clear testing of whether non-photoreceptor-specific genes and proteins are being expressed or not following 4-compound treatment. In all cases sampling sizes should be increased. Further, additional markers for retinal cell-type (e.g., additional markers for non-AII amacrine, horizontal, and bipolar cells, which may exhibit stage- or state-dependent expression patterns of the few chosen markers) should be tested.

In parallel to these concerns, there does not appear to be any discussion of the chosen markers with regards to cell-type specificity. This should be explained.

F1E is not convincing. More images needed, with arrows pointing to morphological changes that support the authors' interpretation.

F1F - data do not appear as though they would pass a normality test; a t-test is thus inadequate for testing statistical differences. Separately, sampling sizes are simply too small to provide meaningful interpretation. This is especially true in testing whether non-photoreceptor-specific expression is occurring or not (e.g., RBPMS expression).

F1H - Similar concerns to F1F.

Line 238 - Change to "cells that express photoreceptor-specific mRNAs or proteins" or some such

Line 252 - Change "On one hand," to "On the other hand,"

F2b - Images are too dim to discern rho expression. Please submit higher quality figures.

F2b-D - Please provide images of all 3 of the entire retina sections for each cohort since by the nature of the outcome measure "rho-expressing retinal region," the selection of the regions is observer-specific and may be biased. Providing complete images will all the reviewer to independently assess outcomes.

F2H, K, I - The observable positively IHC labeled cells are exceedingly few and the images are very limited in the amount of tissue shown, making much of the data of F2H,K,I unconvincing. The small sampling size (N = 3) exacerbates this problem. Sampling sizes should be increased and additional images showing more tissue should be provided.

F2J - Small sampling size and variability of responses precludes meaningful interpretation and, separately, suggest incorrect statistical test was chosen.

FS7 - In this experiment, Td-Tomato-expressing MC were isolated from one set of mice and subretinally injected into another set of mice. Recipient mice were then intravitreally-injected with either DMSO or the 4-compound treatment. It is stated that images in FS7B indicate that only 4-compound-treated eyes showed co-labeling for Td-Tomato and Rho, which would suggest conversation of MC to Rho-expressing phenotype. This is not entirely supported by the data images. There are no Td-Tomato-positive cells at all in the DMSO-treated example; shouldn't there should be some Td-Tomato signal? Second and of concern, this experiment appears to have not been repeated. That is, this appears to be a sampling size of N = 1. This is uninterpretable. Replicates are needed. Finally, if replication robustly supports the current interpretation presented by the authors, the data should not be presented in Supplement, but rather in the body of the manuscript.

Line 369 and 379 (and possibly elsewhere) - "dysfunction" should be "function"

Line 336 and elsewhere - Throughout, there is conflation of the phrase "cell regeneration" with the simple appearance of rhodopsin, possibly in over-expression from rods remaining after MNU-induced loss or in mid-stage RD in rd10 retina, possibly from non-rod cells expressing rhodopsin. Either way, that is not "regeneration" and calling it such is an extreme over-interpretation of modest results. This phrase should be removed in nearly all instances. The statement in line 341 "These data suggest that the Rho-positive cells may be newly formed" is closer to a supported interpretation as it refers to the observation that Rho-expressing cells also stain positive for BrdU.

6. PLOS authors have the option to publish the peer review history of their article (what does this mean?). If published, this will include your full peer review and any attached files.

Reviewer #1: No

Reviewer #2: No

---

## [Author Response · Author response to Decision Letter 0]

20 Dec 2022

A point-by-point response to the issues raised by Editorial office and Reviewers

We would like to thank the editorial office and reviewers for their insightful comments that contributed to the improvement of the manuscript. We have addressed the reviewers’ comments in a point-by-point fashion and have summarized the specific changes as highlighted text in the revised manuscript.

Journal Requirements:

Response: We have referred to the URLs provided by the editorial office and subsequently improved our manuscript to meet PLOS ONE's style requirements.

2. As part of your revision, please complete and submit a copy of the Full ARRIVE 2.0 Guidelines checklist, a document that aims to improve experimental reporting and reproducibility of animal studies for purposes of post-publication data analysis and reproducibility: https://arriveguidelines.org/sites/arrive/files/Author%20Checklist%20-%20Full.pdf (PDF). Please include your completed checklist as a Supporting Information file. Note that if your paper is accepted for publication, this checklist will be published as part of your article

Response: We have duly submitted the Full ARRIVE 2.0 Guidelines checklist as a Supporting Information file.

Response: Thanks for this comment. We have now added information on the grant received from Senju Pharmaceutical Co. The text in the "Funding Information Section" is as follows; This study was supported by JSPS KAKENHI [grants number JP18H02956 and JP21H03094] given to KH Sonoda. Research funding (FAJK310119) was acquired from Senju Pharmaceutical Co., Ltd., given to M Arima, Y Murakami, and KH Sonoda.

“The authors thank Iori Wada, Mitsuhiro Kurata, Masayo Eto, and Fumiyo Morikawa (Kyushu University) for the technical assistance. This study was supported by JSPS KAKENHI [grants number JP18H02956 and JP21H03094] given to KH Sonoda. Research funding was acquired from Senju Pharmaceutical Co., Ltd., given to M Arima, Y Murakami, and KH Sonoda. The funders had no role in the study design, data collection and analysis, decision to publish, or preparation of the manuscript. The authors also thank Editage (www.editage.com) for English language editing.”

“This study was supported by JSPS KAKENHI [grants number JP18H02956 and JP21H03094] given to KH Sonoda. Research funding was acquired from Senju Pharmaceutical Co., Ltd., given to M Arima, Y Murakami, and KH Sonoda. The funders had no role in study design, data collection and analysis, decision to publish, or preparation of the manuscript.”

Response: We appreciate the suggestion from the editorial office. We have removed the funding information from the Acknowledgments Section and added the amended statements to the cover letter.

Response: Thanks for the comment. In accordance with PLOS ONE policy, quantitative data obtained from this study were submitted as a Supporting Information file, and the same has been mentioned in the cover letter for updating the Data Availability statement.

Response: Thanks for this comment. In accordance with PLOS ONE policy, we have submitted the original image of Figure 1G as a Supporting Information file.

 

Reviewer #1: 

Major questions:

1) In the MNU model, it is unclear to me why the authors chose to begin the SLCD treatment at the onset of the retinal damage induction rather than waiting for the damage to be complete and then starting treatment. The way the experiment is currently performed, it is very difficult to completely rule out that the SLCD isn't exerting a preservation effect on the existing photoreceptors rather than inducing the differentiation of new photoreceptors. Although the authors tested for apoptosis, it is possible the SLCD combo could prevent other types of cell death for example. The fact that the combo is ineffective in preserving ERG function when a MNU dose with high photoreceptor toxicity is used but slightly effective when a dose that causes incomplete photoreceptor cell death, also suggests that the SLCD combo might protect some photoreceptors from degeneration.

Response: We appreciate the reviewer’s insightful comment. As the reviewer points out, a preservation effect on existing photoreceptors has not been completely ruled out.

We confirmed that Rho-positive cells appeared even when SLCD was administered after photoreceptor cells were eliminated by MNU, but the ERG amplitude was not restored. This suggests that SLCD is capable of producing new Rho-positive cells in vivo. We presented these results on page 14, lines 325–328 and in S6 Fig. Although MC injury from the 2-week MNU-induced retinal injury may have contributed to the failure of Rho-positive cells to appear in sufficient quantities to restore retinal function, these results suggest that the compound treatment would not generate enough rod-like cells to produce a therapeutic effect in patients with end-stage retinal degeneration.

We administered SLCD during the progression of retinal degeneration because we considered that intravitreal administration of the compounds might be an earlier treatment strategy for patients with retinal degeneration than cell transplantation. S6 Fig results show that compounds are less effective in treating patients with end-stage retinal degeneration and that cell transplantation therapy is superior. However, most of the patients with retinal degeneration that we ophthalmologists usually see have preserved visual function. Especially for RP, there is no effective treatment, and we ophthalmologists are not able to save our patients from future vision loss. Compound therapy is less invasive and easier to administer repeatedly than cell or gene therapy, with the additional advantage of being less expensive. Thus, compound therapy development is in demand for early-stage patients with preserved visual function. The clinical problems and the reasons for the timing of compound administration were added on page 3, lines 48–55, page 3–4, lines 63–65, and page 21–22, lines 535–543.

It has been reported that activation of Wnt signaling prevents the reduction of ERG amplitude by suppressing the amount of apoptosis (newly added reference 62). "C" in SLCD promotes Wnt signaling activation, but in the present study, there was no change in the amount of apoptosis, suggesting that visual function was restored through another mechanism. However, as the reviewer pointed out, the possibility that the protection of residual photoreceptor cells contributed to the recovery of retinal function cannot be ruled out. We cannot conclude that the appearance of Rho-positive cells alone promoted ERG recovery, which we have added to page 23-24, lines 590–597 as a limitation of this study.

We have also added the following reference:

“62. Wyse Jackson AC, Cotter TG. The synthetic progesterone Norgestrel is neuroprotective in stressed photoreceptor-like cells and retinal explants, mediating its effects via basic fibroblast growth factor, protein kinase A and glycogen synthase kinase 3β signalling. Eur J Neurosci. 2016;43: 899-911. doi: 10.1111/ejn.13166.”

2) Why didn't the authors perform ERG analysis on their Rd10 mouse model to assess whether there is a functional benefit?

Response: We agree with the reviewer’s comment. The effects of SLCD on ERG in rd10 mice were added in Fig 6. Similar to the MNU-treated mice, treatment of rd10 mice with SLCD suppressed the decrease in ERG amplitude associated with the progression of retinal degeneration. We appended the ERG results to page 19, lines 466–468. 

Other questions and concerns:

3) The authors need to specify the vivarium lighting conditions in the methods, particularly since degeneration in the Rd10 model is affected by the level of illumination.

Response: We are thankful for this comment. We have mentioned the vivarium lighting conditions. (page 4, lines 81–82).

4) Provide the catalog numbers for the five compounds used in the study

Response: Thank you for this comment. We have added the catalog numbers for the five compounds. (page 5, lines 109–112).

5) Intravitreal injection: the authors need to state the time of day when the compounds were injected as well as the vehicle and the solution pH.

Response: Thanks for the comment. We have stated the time of day of intravitreal injection of compounds, vehicle, and the solution pH. (page 6, lines 116-118).

6) Catalog numbers and dilution factors should be given for all antibodies used in the study.

Response: Thanks for this comment. We have addressed catalog numbers and dilution factors for all antibodies. (page 7–8, lines 142–146, 160–164, and 173–177).

7) I believe the "T" in TGF-b most commonly refers to "transforming", not "tumor"

Response: We are thankful to you for pointing out this error. We have rectified it accordingly. (page 2, line 28 and page 11, line 264).

8) Page 10, line 251: which cone opsin transcript was analyzed.

Response: We analyzed the short-wavelength opsin. Now, we have mentioned the same in the manuscript. (page 12, line 272 and S1 Table)

9) In the figures, the authors might consider referring to their combination as "SLCD" instead of "4 compounds" in order to make it clear which combination is being used in the subsequent studies.

Response: Thanks for pointing this out. We have changed "4 compounds" to "SLCD" in the text and all figures.

10) Can the authors speculate why inclusion of the "Y" compound (ROCK inhibitor) in the treatment combo is so antagonistic to the effects of the SLCD components?

Response: We thank the reviewer for raising a significant concern. It is known that ROCK inhibition contributes to increased survival of ES cells and axon elongation of hippocampal neurons and also protects Müller cells from cellular damage caused by oxidative stress and hypoxia in the retina. We do not know why increasing Y in vitro inhibits differentiation into Rho-positive cells, but we would like to verify this using a comprehensive analytical approach in the future. We have described the effect of "Y" on page 21, lines 521–525, and added the relevant references.

11) Did the authors ever look past Day 7 in their MNU model to see how durable the expression changes are?

Response: We agree with the reviewer’s assessment. Following the reviewer's suggestion, we tested whether Rho expression persisted after compound administration was discontinued. As shown in S9 Fig, we observed significant residual Rho-positive cells on day 14 after SLCD administration was discontinued by day 6. These results suggest that short-term administration of the compound may maintain some long-term recovery of retinal function. The duration of administration is a very important item in drug discovery, and we are very grateful to the reviewer for pointing this out. The results regarding the persistence of Rho-positive cell expression have been added to page 15, lines 364–366, and the discussion has been added to page 24, lines 598–601. 

12) Retinal sections in the main figures need to be labeled properly with retinal layers marked.

Response: Thanks to the reviewer for the comment regarding the figures in the manuscript. We have labeled the retinal layers properly in the figures (Figs 2, 3, and 5).

13) In Figure 3, it would be helpful to indicate what the 75 and 30 mg/kg refers to directly on the figure.

Response: Thanks for the comment. We depicted the 75 mg/kg (MNU) results with a red line and the 30 mg/kg (MNU) results with a blue line in Fig 4.

Reviewer #2: 

However, execution of individual experiments is flawed to the point that interpretation of data is equivocal at best. In nearly all experiments, the sampling size is inappropriately small (N = 3 or 4). Culture experiments are easily repeated, neither mouse strain is especially difficult to breed, and the experimental manipulations are standard to any RD lab. This lack of replicates resulted in highly variable observations within cohorts and in many cases, uninterpretable outcomes. Sampling sizes need to be increased. Similarly, in several instances it appears that the data would not pass a normality test, yet statistical tests that require such were chosen. Nonparametric tests should be used, or better yet, replicate numbers for nearly all experiments simply need to be increased.

Response: We agree with the reviewer’s comment. We have now increased the sample size to 5 or more for all experiments for which we have performed statistical analyses and compared the results using a nonparametric test (Wilcoxon rank sum test).

Overall the experimental design and the data do not exclude the possibility that the 4-compound treatment isn't stimulating Rho expression from vestigial rods in the in vivo models. These deficiencies adds to difficulties in data interpretation.

Similarly, the use of the word or concept of "regeneration" is simply not supported by the data, and critically, could not be supported by the existing experimental design. It may be that the 4-compound treatment induces MCs to express a rod-like phenotype, but even that would not be "regeneration" of photoreceptor cells.

Response: We appreciate the reviewer’s insightful comment. SLCD treatment was performed during the process of photoreceptor cell loss, therefore, it was not possible to distinguish whether Rho expression appeared due to MC differentiation or stimulation of the remaining photoreceptor cells.

Therefore, after photoreceptor cells were eliminated by MNU, SLCD was administered (S6 Fig). The number of Rho-positive cells was higher in the SLCD group than in the DMSO group, suggesting that MC-derived Rho-positive cells may have appeared in vivo. However, Rho-positive cells alone, which would have differentiated from MCs, could not promote recovery of ERG amplitude. These results show that cell transplantation therapy is superior for patients with end-stage retinal degeneration who have lost visual function.

We conducted this study in the hope that SLCD treatment would provide early treatment for patients with retinal degeneration. Most of the patients with retinal degeneration that we ophthalmologists see on a regular basis have preserved visual function. There is no effective treatment, especially for RP, and we ophthalmologists cannot save our patients from future vision loss. Compound therapy has the advantage of being less invasive, less expensive and easier to administer repeatedly than cellular or gene therapy. In short, compound therapy needs to be developed for early-stage patients with preserved visual function. Therefore, we tested the effects of SLCD using an animal model of progressive retinal degeneration. The clinical problems and the reasons for the timing of compound administration were added on page 3, lines 48–55, page 3-4, lines 63–65 and page 21-22, lines 535-543.

The present study does not prove that the appearance of MC-derived Rho-positive cells is directly linked to improved ERG amplitude. As the reviewer pointed out, it is possible that the protective effect on residual photoreceptor cells may have restored visual loss.

We removed the word "regeneration" because we considered that there is not enough evidence to call the intravitreal injection of SLCD a retinal regeneration therapy. In addition, the possibility that protection of residual photoreceptor cells contributes to the recovery of visual function was added to page 23-24, lines 596–602 as a limitation of this study.

The rd10 model should be more-fully tested by assessing effects of 4-compound treatment on retinal function as measured by ERG (as was attempted in the MNU model).

Response: We agree with the reviewer's comments. As shown in Fig 6, treatment of rd10 mice with SLCD suppressed the decrease in ERG amplitude associated with the progression of retinal degeneration. We appended the ERG results to page 19, lines 466–468.

The following are comments to specific datasets ("F" = "figure") and text by line number; note that several of these are of considerable concern:

Line 34 and many other instances - a cocktail of four (4) compounds is stipulated, but the culture experiments apparently used a cocktail with five (5) compounds. Please explain the discrepancy. Also, please ensure that 4 or 5 is correctly stipulated in all instances in text and figures.

Lines 99/100 compared to Line 108 - It appears that MC cultures were tested with a five (5) compound cocktail of SB431542, LDN193189, CHIR99021, DAPT, and Y-27632 but that the in vivo models were tested with a cocktail of just four (4) compounds, SB431542, LDN193189, CHIR99021, and DAPT, but not Y-27632. Please explain why Y-27632 was not included in in vivo treatments. Also, please discuss the implications to interpretation of outcomes that the culture models were not treated with the same cocktail as the in vivo models.

Response: We appreciate the reviewer’s comment. In our research, we first tested five compounds (SB431542, LDN193189, CHIR99021, DAPT, and Y-27632) in vitro to determine the combination of compounds most likely to differentiate from muller cells into photoreceptor cells (Fig1B). As a result, we found that the administration of 4 compounds (SB431542, LDN193189, CHIR99021, and DAPT) increased Rhodopsin expression the most in the cultured cells, and we proceeded with the experiments in vitro (Fig 1C–H) and in vivo (Fig 2~6) using the 4 selected compounds (SB431542, LDN193189, CHIR99021, and DAPT).

We decided that the phrase "four compounds" can be misleading and decided to describe the treatment with a combination of the four compounds selected via in vitro screening as "SLCD treatment".

ROCK inhibition improves ES cell survival, contributes to axon elongation of hippocampal neurons, and protects Müller cells from cell damage caused by oxidative stress and hypoxia in the retina. The reason why the cocktail with added "Y" suppresses differentiation into Rho-positive cells is unknown, but we would like to verify it by exhaustive analytical methods in the future. We have described the effect of "Y" on page 21, lines 521–525, and added references.

F1 - The small sampling sizes and very limited marker choices do not allow for clear testing of whether non-photoreceptor-specific genes and proteins are being expressed or not following 4-compound treatment. In all cases sampling sizes should be increased. Further, additional markers for retinal cell-type (e.g., additional markers for non-AII amacrine, horizontal, and bipolar cells, which may exhibit stage- or state-dependent expression patterns of the few chosen markers) should be tested.

Response: We thank the reviewer for their comments. We increased our sample size to 5 or more and have examined with additional retinal cell-type specific markers (Amacrine cell; Meis2 and Tcf4, Bipolar cell; PCP2, Horizontal cell; Snap25). The results are shown in S4 Fig. 

In parallel to these concerns, there does not appear to be any discussion of the chosen markers with regards to cell-type specificity. This should be explained.

Response: We appreciate the reviewer’s comment. Retinal neuron markers were selected based on previous literature (newly added references 22–28). Neuronal cell types and their corresponding markers are summarized in S2 Table.

S2 Table. Lists of markers for the identification of retinal cells in qPCR.

We have now added the following references:

22. Rodriguez AR, de Sevilla Müller LP, Brecha NC. The RNA binding protein RBPMS is a selective marker of ganglion cells in the mammalian retina. J Comp Neurol. 2014;15: 1411-43. doi: 10.1002/cne.23521.

23. Pérez de Sevilla Müller L, Azar SS, de Los Santos J, Brecha NC. Prox1 Is a Marker for AII Amacrine Cells in the Mouse Retina. Front Neuroanat. 2017;5;11:39. doi: 10.3389/fnana.2017.00039.

24. Yan W, Laboulaye MA, Tran NM, Whitney IE, Benhar I, Sanes JR. Mouse Retinal Cell Atlas: Molecular Identification of over Sixty Amacrine Cell Types. J Neurosci. 2020;40: 5177-5195. doi: 10.1523/JNEUROSCI.0471-20.2020. Epub 2020 May 26.

25. Elshatory Y, Deng M, Xie X, Gan L. Expression of the LIM-homeodomain protein Isl1 in the developing and mature mouse retina. J Comp Neurol. 2007;1: 182-97. doi: 10.1002/cne.21390.

26. Korecki AJ, Cueva-Vargas JL, Fornes O, Agostinone J, Farkas RA, Hickmott JW, et al. Human MiniPromoters for ocular-rAAV expression in ON bipolar, cone, corneal, endothelial, Müller glial, and PAX6 cells. Gene Ther. 2021;28: 351-372. doi: 10.1038/s41434-021-00227-z. Epub 2021 Feb 2.

27. Munro K, Rees S, O'Dowd R, Tolcos M. Developmental profile of erythropoietin and its receptor in guinea-pig retina. Cell Tissue Res. 2009;336: 21-9. doi: 10.1007/s00441-009-0754-5. Epub 2009 Feb 13.

28. Hirano AA, Brandstätter JH, Morgans CW, Brecha NC. SNAP25 expression in mammalian retinal horizontal cells. J Comp Neurol. 2011;519: 972-88. doi: 10.1002/cne.22562.

F1E is not convincing. More images needed, with arrows pointing to morphological changes that support the authors' interpretation.

Response: Thanks for your valuable comment. We have now presented a clearer image and shown arrowheads on a neurite-like structure of the processes (Fig 1E). 

F1F - data do not appear as though they would pass a normality test; a t-test is thus inadequate for testing statistical differences. Separately, sampling sizes are simply too small to provide meaningful interpretation. This is especially true in testing whether non-photoreceptor-specific expression is occurring or not (e.g., RBPMS expression).

F1H - Similar concerns to F1F.

Response: We agree with the reviewer’s assessment. For all experiments for which statistical analyses were performed, the sample size was increased to 5 or more, and the groups were compared using nonparametric tests (Wilcoxon rank sum test).

Line 238 - Change to "cells that express photoreceptor-specific mRNAs or proteins" or some such

Line 252 - Change "On one hand," to "On the other hand,"

Response: We appreciate the reviewer’s suggestions. We have now modified these phrases accordingly (page 11, lines 257–258, and page 12, line 273).

F2b - Images are too dim to discern rho expression. Please submit higher quality figures.

Response: Thanks for the comment. We have replaced the images with higher-quality ones, showing the Rho expression lines with arrowheads (Fig 2B).

F2b-D - Please provide images of all 3 of the entire retina sections for each cohort since by the nature of the outcome measure "rho-expressing retinal region," the selection of the regions is observer-specific and may be biased. Providing complete images will all the reviewer to independently assess outcomes.

Response: Thanks for the comment. Fig 2B includes the entire retina. Images of nuclear staining were added, and the words "the front edge" and white dotted lines were written in the figure to help the reader understand the extent of the neural retina.

F2H, K, I - The observable positively IHC labeled cells are exceedingly few and the images are very limited in the amount of tissue shown, making much of the data of F2H,K,I unconvincing. The small sampling size (N = 3) exacerbates this problem. Sampling sizes should be increased and additional images showing more tissue should be provided.

Response: We agree with the reviewer’s comment. We have increased our sample size to 5, and provided additional images. (Fig 2H, K and I) 

F2J - Small sampling size and variability of responses precludes meaningful interpretation and, separately, suggest incorrect statistical test was chosen.

Response: We appreciate the reviewer’s comment. We used a sample size of 5 or more for all experiments for which we performed statistical analyses and compared the results using a nonparametric test (Wilcoxon rank sum test).

FS7 - In this experiment, Td-Tomato-expressing MC were isolated from one set of mice and subretinally injected into another set of mice. Recipient mice were then intravitreally-injected with either DMSO or the 4-compound treatment. It is stated that images in FS7B indicate that only 4-compound-treated eyes showed co-labeling for Td-Tomato and Rho, which would suggest conversation of MC to Rho-expressing phenotype. This is not entirely supported by the data images. There are no Td-Tomato-positive cells at all in the DMSO-treated example; shouldn't there should be some Td-Tomato signal? Second and of concern, this experiment appears to have not been repeated. That is, this appears to be a sampling size of N = 1. This is uninterpretable. Replicates are needed. Finally, if replication robustly supports the current interpretation presented by the authors, the data should not be presented in Supplement, but rather in the body of the manuscript.

Response: We agree with the reviewer’s comment. We increased our sample size to 5 and provided an image that showed subretinally injected Td-Tomato-positive cells, which were not expressed rhodopsin. These results were added to Fig 3 (page 16-17, lines 393-407) instead of S7 Figure.

Line 369 and 379 (and possibly elsewhere) - "dysfunction" should be "function"

Response: We appreciate the reviewer’s cautions. We have modified these points accordingly (page 17, lines 409,414 and 418). 

Line 336 and elsewhere - Throughout, there is conflation of the phrase "cell regeneration" with the simple appearance of rhodopsin, possibly in over-expression from rods remaining after MNU-induced loss or in mid-stage RD in rd10 retina, possibly from non-rod cells expressing rhodopsin. Either way, that is not "regeneration" and calling it such is an extreme over-interpretation of modest results. This phrase should be removed in nearly all instances. The statement in line 341 "These data suggest that the Rho-positive cells may be newly formed" is closer to a supported interpretation as it refers to the observation that Rho-expressing cells also stain positive for BrdU.

Response: We appreciate the reviewer’s insightful comment. As the reviewer pointed out, there is not enough evidence to call the intravitreal injection of SLCD a retinal regeneration therapy. Therefore, we have replaced the term “regeneration” with “production” (page 22, line 552). 

We thank the reviewers for the time and effort invested in reviewing the manuscript. We truly appreciate the constructive comments that have contributed significantly to the improvement of the manuscript.

---

## [Decision Letter · Decision Letter 1]

23 Jan 2023

PONE-D-22-16150R1Rhodopsin-positive cell production by intravitreal injection of

small molecule compounds in mouse models of retinal degenerationPLOS ONE

Dear Dr. Arima,

Thank you for submitting your manuscript to PLOS ONE. After careful consideration, we invite you to submit a revised version of the manuscript that addresses the minor points raised during the review process.

Please submit your revised manuscript by Mar 09 2023 11:59PM. If you will need more time than this to complete your revisions, please reply to this message or contact the journal office at plosone@plos.org. Please include the following items when submitting your revised manuscript:A rebuttal letter that responds to each point raised by the academic editor and reviewer(s). You should upload this letter as a separate file labeled 'Response to Reviewers'.A marked-up copy of your manuscript that highlights changes made to the original version. You should upload this as a separate file labeled 'Revised Manuscript with Track Changes'.An unmarked version of your revised paper without tracked changes. You should upload this as a separate file labeled 'Manuscript'.If applicable, we recommend that you deposit your laboratory protocols in protocols.io to enhance the reproducibility of your results. Protocols.io assigns your protocol its own identifier (DOI) so that it can be cited independently in the future. For instructions see: https://journals.plos.org/plosone/s/submission-guidelines#loc-laboratory-protocols. Additionally, PLOS ONE offers an option for publishing peer-reviewed Lab Protocol articles, which describe protocols hosted on protocols.io. Read more information on sharing protocols at https://plos.org/protocols?utm_medium=editorial-email&utm_source=authorletters&utm_campaign=protocols.

We look forward to receiving your revised manuscript.

Kind regards,

Anand Swaroop

Academic Editor

PLOS ONE

Journal Requirements:

Reviewers' comments:

Reviewer's Responses to Questions

**Comments to the Author**

1. If the authors have adequately addressed your comments raised in a previous round of review and you feel that this manuscript is now acceptable for publication, you may indicate that here to bypass the “Comments to the Author” section, enter your conflict of interest statement in the “Confidential to Editor” section, and submit your "Accept" recommendation.

Reviewer #1: All comments have been addressed

Reviewer #2: (No Response)

2. Is the manuscript technically sound, and do the data support the conclusions?

Reviewer #1: Yes

Reviewer #2: Partly

3. Has the statistical analysis been performed appropriately and rigorously? 

Reviewer #1: Yes

Reviewer #2: Yes

4. Have the authors made all data underlying the findings in their manuscript fully available?

Reviewer #1: Yes

Reviewer #2: Yes

5. Is the manuscript presented in an intelligible fashion and written in standard English?

Reviewer #1: Yes

Reviewer #2: Yes

6. Review Comments to the Author

Reviewer #1: The authors have thoroughly addressed my comments and concerns and have appropriately tempered the conclusions of the manuscript in light of remaining uncertainties regarding the action of their combination therapy.

Reviewer #2: It is appreciated that the authors made a strong and good-faith effort to address reviewer concerns, particularly with addition of new experiments and increased sampling sizes of original experiments. The remaining issue is the use of the word "regeneration" in a manner that is confusing. The authors state in their response that they removed the word "regeneration," yet it appears throughout the manuscript.

As noted previously the use of the word or concept of "regeneration" as generally understood is not supported by the data, and critically, could not be supported by the existing experimental design. The authors are asked to consider that the word "regeneration" does not appear in the body of the text of the studies that they reference as examples of "central nerve regeneration" (lines 57-61, references 9-12). The authors' data do support that their test treatment resulted in observation of "Rhodopsin-positive cell production," as correctly stated in the title of the manuscript. The authors are asked to consider using similar phrases in the body of the text rather than "regeneration." Alternatively, the authors could very precisely and thoroughly describe in the Introduction exactly what they mean by the word "regeneration" and in the Discussion describe how their data support the existence of that exactly defined outcome. This would allow the reader to understand what the authors mean when they use the word "regeneration."

7. PLOS authors have the option to publish the peer review history of their article (what does this mean?). If published, this will include your full peer review and any attached files.

Reviewer #1: No

Reviewer #2: No

---

## [Author Response · Author response to Decision Letter 1]

6 Feb 2023

Reviewer #2: 

It is appreciated that the authors made a strong and good-faith effort to address reviewer concerns, particularly with addition of new experiments and increased sampling sizes of original experiments. The remaining issue is the use of the word "regeneration" in a manner that is confusing. The authors state in their response that they removed the word "regeneration," yet it appears throughout the manuscript.

Other questions and concerns:

As noted previously the use of the word or concept of "regeneration" as generally understood is not supported by the data, and critically, could not be supported by the existing experimental design. The authors are asked to consider that the word "regeneration" does not appear in the body of the text of the studies that they reference as examples of "central nerve regeneration" (lines 57-61, references 9-12). The authors' data do support that their test treatment resulted in observation of "Rhodopsin-positive cell production," as correctly stated in the title of the manuscript. The authors are asked to consider using similar phrases in the body of the text rather than "regeneration." Alternatively, the authors could very precisely and thoroughly describe in the Introduction exactly what they mean by the word "regeneration" and in the Discussion describe how their data support the existence of that exactly defined outcome. This would allow the reader to understand what the authors mean when they use the word "regeneration."

Response: We thank the reviewer for this comment, as he/she raises an important issue. In our experimental design, SLCD treatment was performed during the process of photoreceptor cell loss; therefore, it was not possible to distinguish whether Rho expression appeared due to MC differentiation or stimulation of the remaining photoreceptor cells.

In the previous revision, we avoided the use of the word “regeneration” relating to our experimental results. However, we agree with the reviewer that the word “regeneration”, when found throughout the body of the text is misleading, as if this study leads to a conclusion that shows neural regeneration.

Thus, as per the reviewer’s suggestion, we have now removed the words “regeneration” and “regenerative” in the text and modified the manuscript as shown in the box below.

---

## [Editor Report · Decision Letter 2]

9 Feb 2023

Rhodopsin-positive cell production by intravitreal injection of

small molecule compounds in mouse models of retinal degeneration

PONE-D-22-16150R2

Dear Dr. Arima,

We’re pleased to inform you that your manuscript has been judged scientifically suitable for publication and will be formally accepted for publication once it meets all outstanding technical requirements.

Kind regards,

Anand Swaroop

Academic Editor

PLOS ONE
---

## [Editor Report · Acceptance letter]

13 Feb 2023

PONE-D-22-16150R2 

Rhodopsin-positive cell production by intravitreal injection of small molecule compounds in mouse models of retinal degeneration 

Dear Dr. Arima:

I'm pleased to inform you that your manuscript has been deemed suitable for publication in PLOS ONE. Congratulations! Your manuscript is now with our production department. 

Kind regards, 

on behalf of

Dr. Anand Swaroop 

Academic Editor

PLOS ONE